# *Kcnq* (Kv7) channels exhibit frequency-dependent responses via partial inductor-like gating dynamics

Yuta Eguchi [1,2], Yuki Kuwano[3], Satoshi Okada [2], Hiroyuki Morino [3] & Kouichi Hashimoto [1] ✉

Kcnq channels are low-threshold voltage-dependent K$^+$ channels that generate M-currents, which regulate the peri-threshold membrane potential. Kcnq channels reportedly participate in band-pass frequency responses (i.e., resonance), but it remains largely unclear how they contribute to generating resonance. We examined resonance in HEK293 cells expressing *mouse Kcnq2* and *Kcnq3* (*Kcnq2/3*) using whole-cell recording. *Kcnq2/3*-expressing cells generated resonance-like frequency-dependent responses. Kcnh7 channels displayed a rapid opposing conductance change followed by slow activation in response to a depolarizing voltage step, properties thought to be necessary for inductor-like activity. However, Kcnq2/3 channels exhibited only slow activation. The lack of an opposing conductance change was caused by the absence of rapid Kcnq2/3 channel inactivation. These data suggest that core ion channel characteristics that cause resonance-like frequency responses are not uniform among ion channels. The opposing conductance change is not necessary for resonance-like frequency responses but is crucial for fine frequency tuning and oscillation.

The activation of various G protein-coupled receptors, including muscarinic receptors, depolarizes certain neurons by suppressing the activity of a low-threshold, non-inactivating K$^+$ conductance called 'M-current'[1]. The M-current is mediated by *KCNQ* voltage-dependent K$^+$ channel subtypes (*Kcnq2* [Kv7.2], *Kcnq3* [Kv7.3], *Kcnq4* [Kv7.4], and *Kcnq5* [Kv7.5])[2,3]. KCNQ channels exhibit unique biophysical properties including low-threshold activation near the resting membrane potential, slow activation kinetics, and little inactivation. These properties enable KCNQ channels to regulate the peri-threshold membrane potential, contributing to the stability and responsiveness of neuronal firing[4–8]. Some pathogenic mutations in *KCNQ2* and *KCNQ3* are associated with neonatal onset epilepsy. For example, certain loss-of-function (LOF) missense mutations are associated with self-limited (familial) neonatal epilepsy (SL[F]NE; previously known as benign familial neonatal convulsions [BFNC])[9,10]. Meanwhile, both LOF and gain-of-function (GOF) mutations in these genes can lead to developmental and epileptic encephalopathy (DEE), characterized by severe epilepsy and neurodevelopmental delays[9,11–13]. These lines of evidence suggest the critical importance of KCNQ channel-mediated regulation of peri-threshold membrane potentials in maintaining proper neuronal excitability.

In addition to regulating the peri-threshold membrane potential, KCNQ channels have also been implicated in mediating the resonant property of neurons. In response to periodic current inputs (e.g., periodic synaptic inputs), neurons exhibit a band-pass filtering property that outputs the input currents with a specific frequency as a larger voltage response, which is known as resonance or the resonant property[14,15]. Resonance has been proposed to arise from the biophysical properties of the plasma membrane, which is conceptually equivalent to a parallel resonant circuit consisting of a resistor, inductor, and capacitor (RLC circuit)[15]. The plasma membrane is biophysically represented as an electrical circuit with a resistor and capacitor in parallel. In living cells, the inductive component is proposed to be mediated by ion channels capable of mimicking inductor-like behavior, which is called resonating conductance[15]. Ion channels consistently reported to contribute resonating conductance include hyperpolarization activated cyclic nucleotide gated potassium (HCN) channels and *KCNH* voltage-dependent K$^+$ channels[14,15]. In addition to resonating conductance, resonance also depends on amplifying conductance, which boosts resonating conductance activity in a voltage-dependent manner[14,15]. Persistent sodium channels are reported as amplifying conductance. T-type voltage-dependent calcium channels have been reported to function as either resonating[15,16] or amplifying conductance[17]. In some neurons, M-current blockers inhibit resonance at relatively depolarized membrane potentials[18–24], and resonance in CA1 pyramidal neurons is impaired in mice expressing a *KCNQ2* subunit with a dominant-negative pore mutation[23]. These reports suggest that KCNQ channels can function as resonating

[1]Department of Neurophysiology, Graduate School of Biomedical and Health Sciences, Hiroshima University, Hiroshima, Japan. [2]Department of Pediatrics, Graduate School of Biomedical and Health Sciences, Hiroshima University, Hiroshima, Japan. [3]Department of Medical Genetics, Tokushima University Graduate School of Biomedical Sciences, Tokushima, Japan. ✉e-mail: hashik@hiroshima-u.ac.jp

conductance, particularly at depolarized potentials[14]. However, the precise mechanism by which they emulate inductor-like properties to drive resonance remains unclear.

In the present study, we examined the functional roles of the Kcnq2 and 3 heteromeric channel (Kcnq2/3) in resonance using whole-cell recording from transfected HEK293 cells. *Kcnq2* and *3* co-assemble into functional heteromeric channels in the central nervous system[2,3]. Our results demonstrated that *Kcnq2/3*-expressing HEK293 cells generated resonance-like frequency-dependent responses, supporting their role as resonating conductance. However, the gating properties of Kcnq2/3 channels were partly different from those of the Kcnh7 channel, which was reported to generate resonance and membrane potential oscillation in the heterologous expression system[25]. Kcnh7 channels displayed rapid opposing conductance changes followed by slow activation and deactivation in response to square voltage steps, which are properties thought to be necessary for inductor-like activity. However, Kcnq2/3 channels exhibited only slow activation and deactivation properties without the opposing conductance change. Computational analysis revealed that the lack of an opposing conductance change was caused by the absence of rapid ion channel inactivation in Kcnq channels. These data suggest that the core ion channel characteristics that cause resonance-like frequency responses are not uniform among ion channels.

## Results

### Kcnq2/3-expressing HEK293 cells show a resonance-like frequency response

To examine the functional roles of Kcnq channels in resonance, we co-transfected *mouse Kcnq2* and *Kcnq3* subunits into HEK293 cells. Whole-cell recordings were conducted from enhanced green fluorescent protein (EGFP)-positive cells and non-transfected control cells (lacking EGFP expression) in the same culture dishes at 1–3 days after transfection. For comparison, in some experiments, we introduced *Kcnh7* into HEK293 cells, which has been experimentally reported to exhibit inductor-like gating properties and generate resonance and membrane potential oscillations[25]. The depolarizing voltage protocol evoked slowly activating outward currents with little inactivation in EGFP-positive, *Kcnq2/3*-expressing cells (Fig. 1a). These currents were activated at a membrane potential near −50 mV (Fig. 1b) and were inhibited by a Kcnq channel blocker, XE991 (20 μM), in an activity-dependent manner (Fig. 1a, c). These electrophysiological characteristics were consistent with previously reported properties of Kcnq2/3 channels[3].

To examine whether Kcnq2/3 channels can generate resonance, we applied an impedance amplitude profile (ZAP) stimulus as described in previous studies[17,25–27]. A chirp current—characterized by a sinusoidal waveform with constant amplitude and a linearly increasing frequency over time (Fig. 1d)—was injected through the recording electrode, and resulting voltage responses were recorded. When these cells exhibited resonance, the voltage response to the ZAP stimulus reaches a peak amplitude at a specific preferred frequency, referred to as the resonant frequency (e.g., as illustrated in the −40 mV panel in Fig. 1i). Correspondingly, the frequency plot of the impedance magnitude (Z-F plot) exhibits a distinct "hump" centered around the resonant frequency (e.g., as shown in the −40 mV panel in Fig. 1j).

In control HEK293 cells, voltage responses to chirp current injections gradually declined as increase in the input current frequency across all tested membrane potentials (Fig. 1e). This decline was relatively shallow, likely because of the low capacitance of HEK293 cells compared with that of neurons[25]. Moreover, the Z-F plot lacked any distinct hump (Fig. 1f), indicating an absence of resonance in control cells. In contrast, *Kcnh7*-expressing cells exhibited enhanced voltage responses to the chirp current injection at approximately 2–3 seconds after stimulus onset (Fig. 1g), corresponding to a frequency of 2–3 Hz, consistent with a previous finding[25]. These enhanced responses were observed at depolarized membrane potentials over −40 mV. The calculated impedance displayed a distinct hump at a resonant frequency of 2.7 ± 0.3 Hz (−40 mV, n = 19) in the Z-F plot (Fig. 1h). In most *Kcnh7*-expressing cells, resonance above −30 mV could not be analyzed because of the emergence of membrane potential oscillations, which are further described later[25].

In *Kcnq2/3*-expressing HEK293 cells, the voltage response to the ZAP stimulus was similarly enhanced at approximately 8–9 seconds after stimulus onset (Fig. 1i, j). The resonant frequency was 8.4 ± 0.5 Hz (n = 55, −40 mV), which was higher than that in *Kcnh7*-expressing cells (Fig. 1k). At hyperpolarized potentials, the ratio of the maximum impedance at the resonant frequency to the impedance at 0.5 Hz (resonant strength) was approximately 1. However, the resonant strength increased at depolarized potentials of −60 mV and above (Fig. 1l), which closely aligns with the activation threshold of KCNQ2/3 channels. The maximum resonant strength exhibited a weak positive correlation with Kcnq2/3 conductance (r = 0.35, P = 0.003, Pearson's correlation coefficient). The resonant strength declined again at the depolarized membrane potential range (over −30 mV) (Fig. 1l), likely because of the saturation of Kcnq2/3 channel activation. These data suggest that Kcnq2/3 channels show resonance-like frequency-dependent responses with a preferred frequency that is higher than that of Kcnh7 channels. In addition to the difference in resonant frequency, the Z-F plot of Kcnq2/3 channels revealed a gentler decline after the peak than that of Kcnh7 channels (Fig. 1m), indicating a broader tuning curve for Kcnq2/3 channels. Furthermore, impedance in the low frequency range was lower for Kcnq2/3 than for Kcnh7 channels, potentially explaining the higher resonant strength observed in Kcnq2/3 channels (see Methods). Collectively, these results suggest that resonances mediated by Kcnq2/3 and Kcnh7 channels exhibit some different characteristics.

### Kcnq2/3 channels repeat activation and deactivation in response to low frequency sinusoidal voltage changes

We next examined the frequency response characteristics of Kcnq2/3 channel gating using voltage clamp recordings. Kcnq2/3 currents in response to sinusoidal voltage clamping—the same amplitude (15 mV) but different frequencies (0.5, 5, 10, or 15 Hz) from a holding potential of −40 mV—were recorded in the presence or absence of XE991. Kcnq2/3 currents were isolated by subtracting currents of XE991-treated cells from those recorded under control conditions. In an electrical circuit, the impedance of an inductor increases with the frequency of the input current. When a sinusoidal voltage is applied to a parallel RLC circuit, the inductor allows greater current flow at low frequencies because of its lower impedance. As the frequency increases, current flow through the inductor diminishes, and instead, the current begins to flow primarily through the capacitor in the parallel resonant circuit. If Kcnq2/3 channels act analogously to an inductor in a resonance circuit, their current should exhibit similar frequency-dependent behavior[17,25].

XE991-sensitive currents, representing Kcnq2/3 channel activity, displayed an overall trend of a frequency-dependent decrease in peak amplitude as the input voltage frequency increased (Fig. 2a–f). However, the positive currents that were observed in the depolarizing phases were larger than the negative currents observed in the hyperpolarizing phases (Fig. 2b, g, h). These data suggest that Kcnq2/3 channels repeat activation in depolarizing phases and deactivation in hyperpolarizing phases during sinusoidal voltage clamping. Such gating behavior likely serves to counteract membrane potential fluctuations; activation during depolarization drives hyperpolarization, while closure during hyperpolarization allows depolarization. This mechanism is particularly effective at low frequencies, where it contributes to the upward slope of the Z-F plot (Fig. 1j). Meanwhile, at high frequencies, Kcnq2/3 channels showed reduced conductance, likely because their gating could not keep pace with the rapid voltage oscillations. Together, these factors contribute to an upward slope in the lower frequency range of the Z-F plot (Fig. 1j). This dynamic behavior of Kcnq2/3 channels contrasts with that of Kcnh7 channels, which generate largely symmetrical positive and negative currents[25]. Thus, these findings suggest that Kcnq2/3 and Kcnh7 channels use a little different underlaying mechanisms to generate resonance.

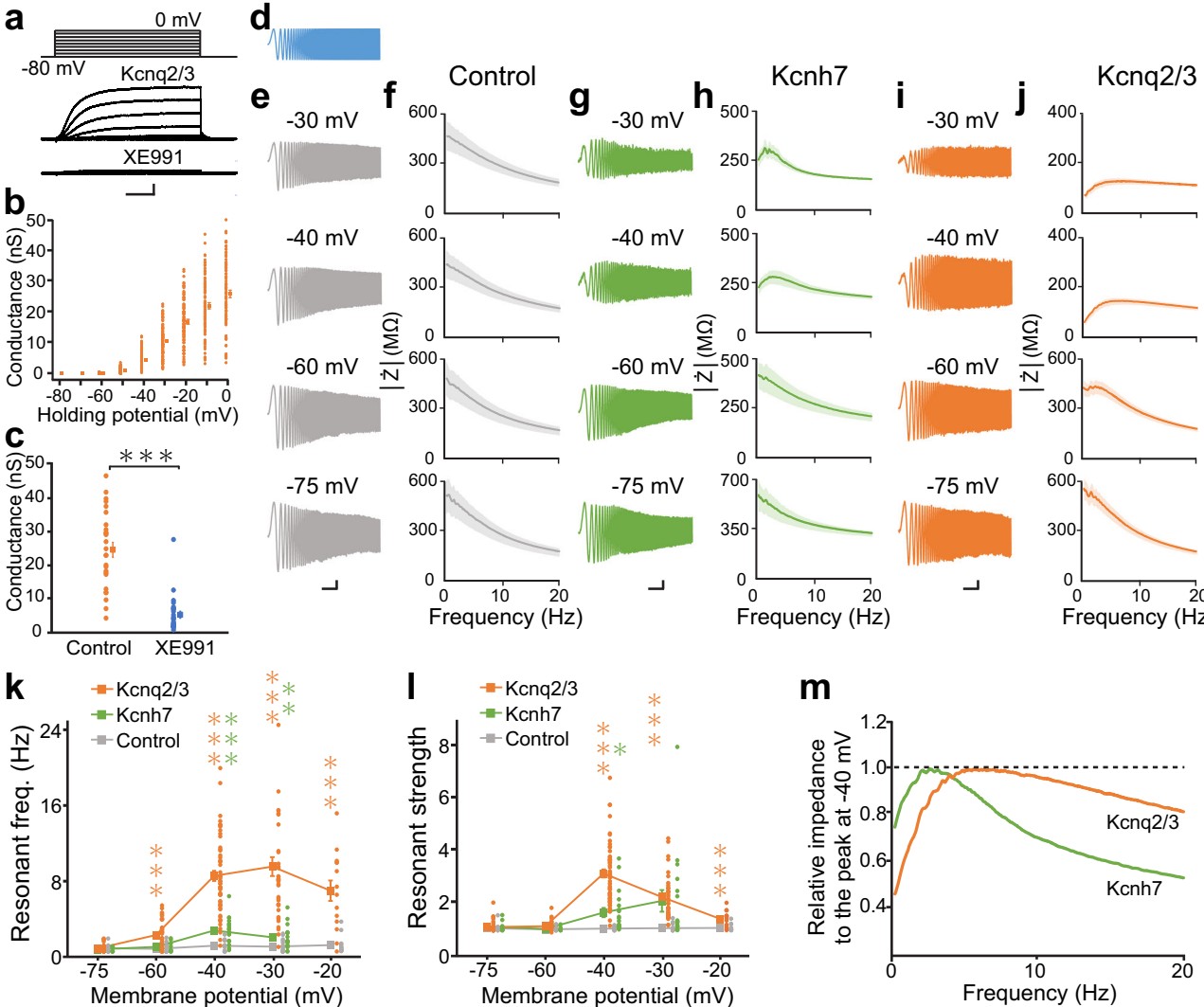

**Fig. 1 | A resonance-like frequency-dependent response is observed in *Kcnq2/3*-expressing HEK293 cells. a** (Top) Depolarizing voltage protocol for activating Kcnq2/3 currents in *Kcnq2/3*-expressing HEK293 cells. Voltage steps of 10 mV for 1 s were applied from −80 to 0 mV of membrane potentials. Representative traces in the absence (middle) and presence (bottom) of XE991 (20 μM). For recordings traces in the presence of XE991, a depolarizing prepulse of over −30 mV for more than 30 s was applied to block Kcnq2/3 channels. **b** Conductance plots of *Kcnq2/3*-expressing HEK293 cells at each holding membrane potential. Conductance was calculated using the theoretical equilibrium potential of $K^+$ (−98 mV). Square symbols and bars represent the mean ± standard error of the mean (SEM; *n* = 88). **c** Pharmacological effects of XE991 on Kcnq2/3 conductance evoked by a membrane potential depolarization to 0 mV. Square symbols and bars represent the mean ± SEM (*n* = 26). ***$P < 0.001$ (paired *t*-test). **d** Waveform of the chirp current consisting of a sinusoidal current with a constant amplitude (50 pA) but a linearly altered frequency ranging from 0–40 Hz over 40 s. **e, g, i** Representative voltage traces in response to the chirp currents at individual membrane potentials in control (**e**), *Kcnh7*-expressing (**g**), and *Kcnq2/3*-expressing (**i**) HEK293 cells. **f, h, j** Averaged

Z-F plot of control (**f**, *n* = 14–16), *Kcnh7*-expressing (**h**, *n* = 9–19), and *Kcnq2/3*-expressing (**j**, *n* = 37–58) HEK293 cells. Shaded areas indicate the mean ± SEM. **k, l** Membrane potential dependence of the resonant frequency (**k**) and resonant strength (**l**) in control (−75 mV, *n* = 16; −60 mV, *n* = 14; −40 mV, *n* = 16; −30 mV, *n* = 17; −20 mV, *n* = 15, gray), *Kcnh7*-expressing (−75 mV, *n* = 9; −60 mV, *n* = 9; −40 mV, *n* = 19; −30 mV, *n* = 17, green), and *Kcnq2/3*-expressing (−75 mV, *n* = 38; −60 mV, *n* = 38; −40 mV, n = 55; −30 mV, *n* = 37; −20 mV, n = 14, orange) HEK293 cells. The resonance could not be examined over −30 mV in *Kcnh7*-expressing cells because of membrane potential oscillations. The resonant frequency and strength at individual membrane potentials were compared with those at -75 mV. ***$P < 0.001$, **$P < 0.01$, *$P < 0.05$ (Kruskal-Wallis test; post-hoc Holm–Šidák test). Small dots represent individual data, and squares with bars present the mean ± SEM. **m** Average Z-F plots of Kcnq2/3 and Kcnh7 channels at −40 mV (those in **h** and **j**) are normalized by peak magnitudes. Kcnq2/3 channels showed a gentler frequency dependence than Kcnh7 channels. In contrast, the impedance at 0.5 Hz was higher for Kcnh7 channels, resulting in lower resonant strength. Scale bars, 0.5 nA, 0.2 s (**a**), 10 mV, 2 s (**e**), 5 mV, 2 s, (**g, i**).

## Kcnq2/3 channels do not show an opposing conductance change

At the onset and offset of rapid current changes, an inductor produces a transient voltage that counteracts the polarity of the applied current. This opposing transient voltage serves to suppress the flow of current through the inductor during these transitions. Consequently, in response to a step voltage change, the inductor current ($I_L$ in Fig. 3a) is initially zero when the voltage is applied and then changes exponentially from the baseline (unlike a step-like current change similar to the applied voltage),

eventually reaching a steady plateau (Fig. 3a–c). We have previously demonstrated that Kcnh7 channels exhibit comparable current dynamics in response to square voltage pulses. This behavior arises from 1) the opposing conductance change and 2) slow activation and deactivation of Kcnh7 channels at the onset and offset of the square voltage clamp[25]. These gating properties effectively suppress the rapid currents induced by abrupt voltage shifts. On the basis of this observation, we hypothesized that these gating properties mimic the transient opposing voltage generated by an inductor.

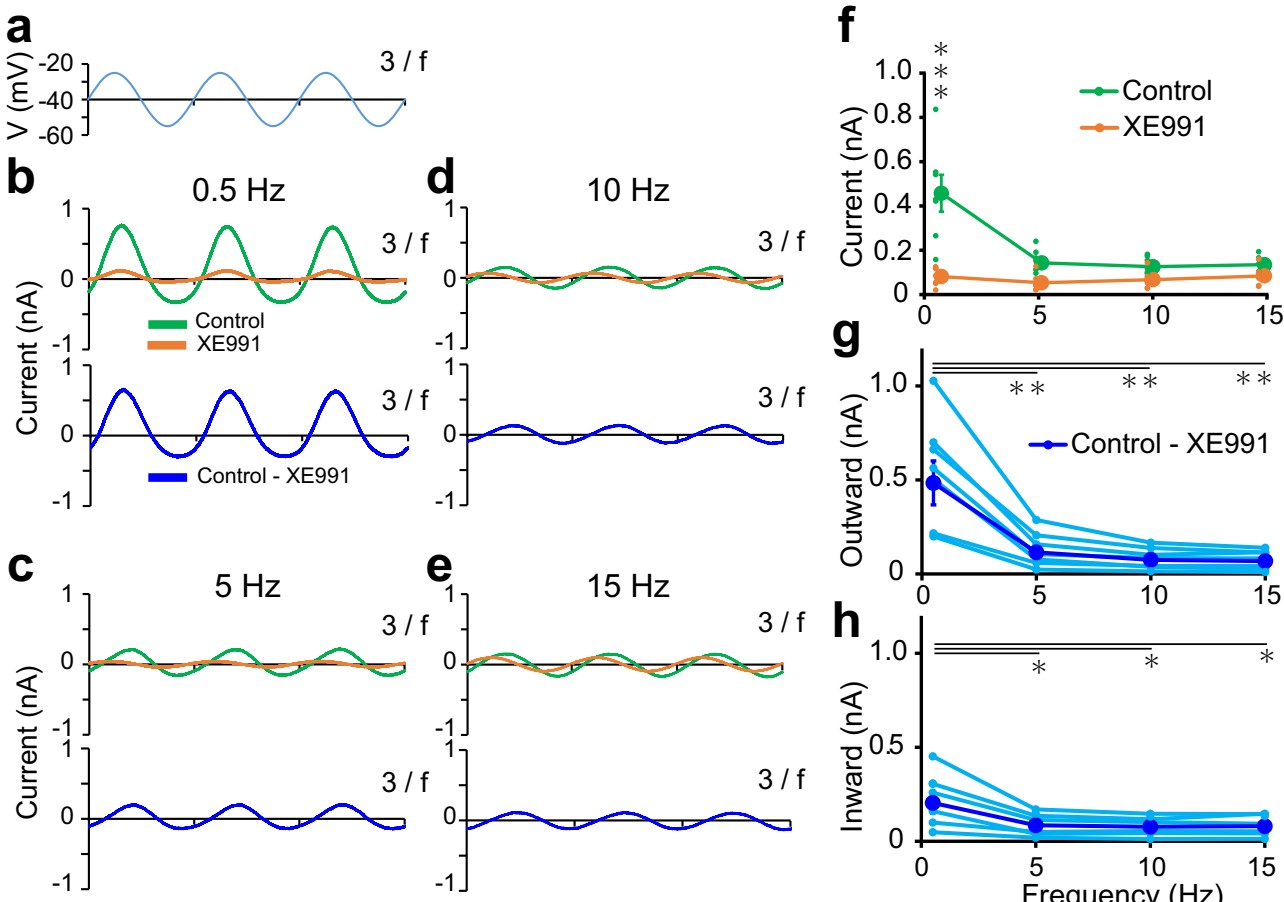

**Fig. 2 | Frequency response characteristics of Kcnq2/3 currents. a** Waveform of the sinusoidal input voltage. **b–e** Representative currents recorded in control (upper traces, green) and XE991-containing (20 μM, upper traces, orange) solutions in response to the sinusoidal input voltage (± 15 mV) at −40 mV in *Kcnq2/3*-expressing HEK293 cells. XE991-sensitive currents (lower traces, blue) were calculated by subtracting the current obtained in the XE991-containing solution from that in the control solution. These recordings were performed with input voltage frequencies of 0.5 (**b**), 5 (**c**), 10 (**d**), and 15 (**e**) Hz. **f** Amplitudes of currents recorded in control (n = 7, green) and XE991-containing (n = 7, orange) solutions plotted against the holding potential frequency. The half-magnitude from the bottom to the top of the sinusoidal membrane current represents the amplitude. Small dots represent individual data and large dots with bars represent the mean ± SEM. ***P < 0.001 (repeated measures two-way ANOVA; post-hoc Holm–Šidák test). **g, h** Frequency dependence of outward (**g**) and inward (**h**) current amplitudes of XE991-sensitive currents (n = 7). Small light-blue dots represent individual data, and large blue dots with bars represent the mean ± SEM. **P < 0.01, *P < 0.05 (repeated measures one-way ANOVA; post-hoc Holm–Šidák test).

To investigate whether Kcnq2/3 channels exhibit similar properties, we analyzed the responses of *Kcnq2/3*-expressing HEK293 cells to square voltage steps. Square voltage steps of ±5 or ±10 mV were applied from a holding potential of −40 mV (Fig. 3d, e), the condition where *Kcnq2/3*-expressing cells exhibited the largest resonant strength (Fig. 1l). Kcnq2/3 currents were isolated by subtracting traces of XE991-treated cells from those recorded under control conditions (Supplementary Fig. 1a, b). For comparison, we conducted parallel experiments using HEK293 cells expressing *Kcnh7* at a holding potential of −30 mV, where *Kcnh7*-expressing cells exhibited the largest resonant strength (Fig. 1l)[25]. In these recordings, E-4031 (10 μM) was used as a Kcnh channel blocker (Supplementary Fig. 1c, d).

As previously reported[25], Kcnh7 currents evoked by +5 mV or +10 mV square voltage steps increased and decreased exponentially from baseline, eventually stabilizing at plateau levels (Fig. 3o, p). This response closely resembled the inductor current ($I_L$) behavior observed during voltage steps (Fig. 3c). We calculated the overall Kcnh7 conductance using the recorded Kcnh7 current and the K⁺ driving force (the equilibrium potential of K⁺: − 98 mV), derived from the intracellular and extracellular K⁺ ion gradient. The Kcnh7 conductance exhibited two response phases to the square depolarizing voltage step: a transient

decrease at the onset of depolarization (arrowheads in Fig. 3q, r) and a subsequent slow increase that was almost saturated at the end of depolarization (Fig. 3q, r). In contrast, the Kcnh7 conductance exhibited a transient increase at depolarization offset and a subsequent slow decrease to the baseline (Fig. 3q–t). At a holding potential of −30 mV, the K⁺ current flowing through Kcnh7 channels increased with depolarization. In this context, such a transient conductance change opposes the current change induced by the voltage step, allowing the current responses to initiate from baseline at the onset of voltage application (Fig. 3o, p). The subsequent slow increase and decrease in conductance suggests voltage-driven slow activation and deactivation of Kcnh7 channels.

At first glance, the Kcnq2/3 current appeared to exhibit similar exponential increases and decreases in response to square voltage pulses (Fig. 3d, e). However, closer examination revealed that these current changes did not originate from distinct steady-state levels. They displayed abrupt stepwise increases and decreases at the onset and offset of depolarization, respectively (Fig. 3f, g). When a stepwise voltage change is applied to ion channels in the steady-state open configuration at −40 mV, the resulting current also exhibits an instantaneous stepwise change, according to Ohm's law. It is likely that a similar mechanism underlies the observed step-like current changes. Further calculation of the Kcnq2/3 conductance revealed

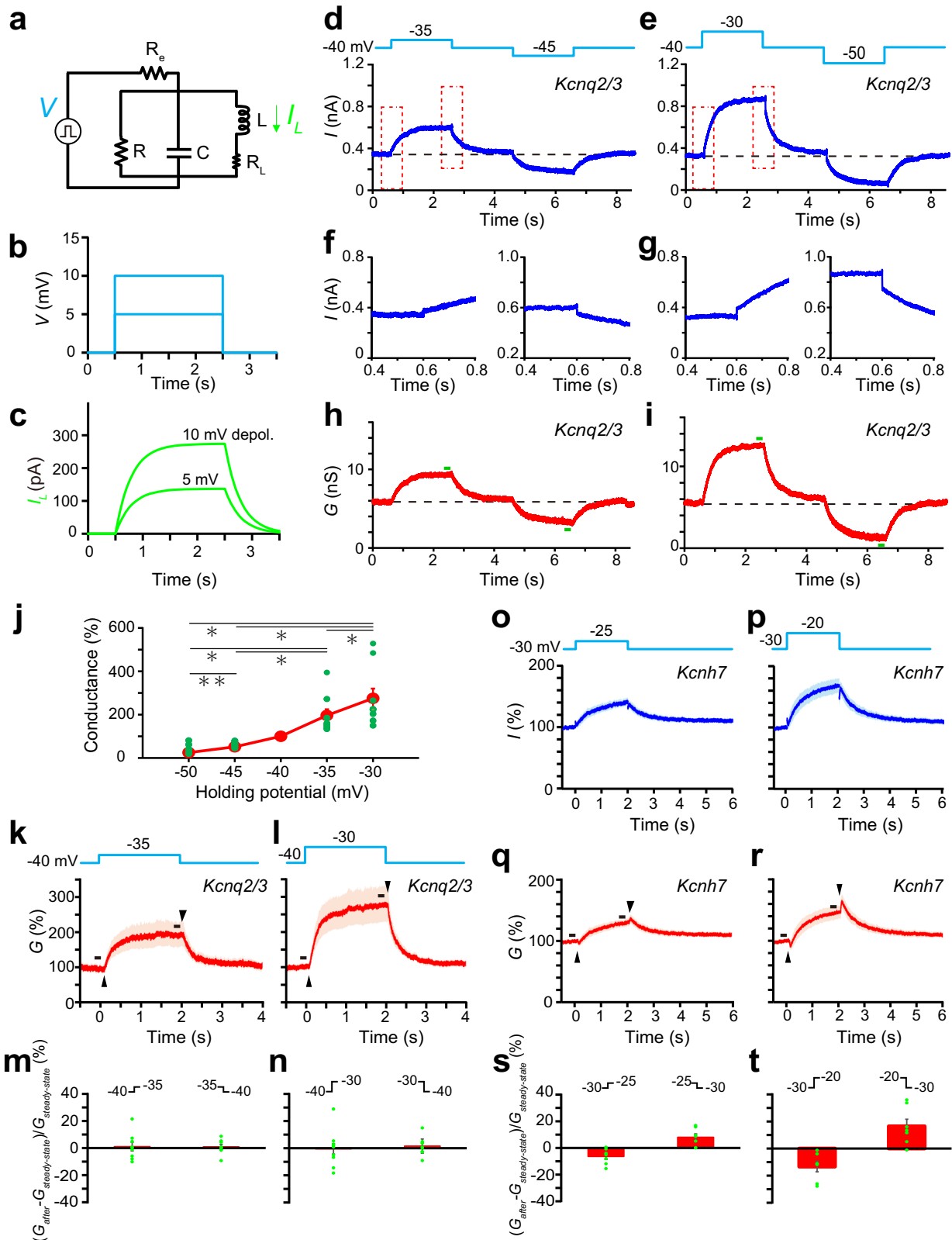

that, unlike Kcnh7, the Kcnq2/3 conductance did not exhibit opposing conductance changes (Fig. 3h, i, k–n). The channels exhibited only a slow increase or decrease to the plateau state (Fig. 3h–j), suggesting voltage-driven slow activation and deactivation of Kcnq2/3 channels. These data suggest that Kcnq2/3 channels lack the capacity to fully mimic the inductor-like performance observed in Kcnh7 channels.

## Kcnq2/3 channels do not generate membrane potential oscillation

We have previously reported that HEK293 cells expressing *Kcnh2* variants or *Kcnh7* show clear membrane potential oscillation at depolarized membrane potentials[25]. To examine the potential rhythmogenetic properties of Kcnq2/3 channels, the membrane potential was changed from −75 mV

**Fig. 3 | Kcnq2/3 current responses to square voltage steps are different from those of Kcnh7 channels. a** Resistor-inductor-capacitor (RLC) circuit with a leaky inductor ($R_L$ [30 MΩ] and $L$ [10 MH]) and resistance of the recording electrode ($Re$ [5 MΩ]). $R = 100$ MΩ, $C = 200$ pF. **b** Square input voltage steps with amplitudes of 5 and 10 mV. **c** Simulated currents flowing through the leaky inductor ($I_L$ in **a**) in response to square input voltage steps. Currents were simulated using LTspice XVII. **d, e** Voltage protocols applied to *Kcnq2/3*-expressing cells (upper traces) and representative Kcnq2/3 currents (lower traces) in response to ±5 (**d**) and ±10 (**e**) mV voltage steps. Kcnq2/3 currents were calculated by subtracting the currents recorded in the presence of XE991 (20 μM) from those recorded in its absence. **f, g** Expanded traces in response to ±5 (**f**) and ±10 (**g**) mV voltage steps in the periods marked by dotted rectangles in **d** or **e**. **h, i** Representative Kcnq2/3 conductance calculated from the data in **d** and **e**. The conductance was calculated by dividing the current amplitude by the driving voltage (holding potential—theoretically calculated K⁺ equilibrium potential [−98 mV]). **j** Steady-state Kcnq2/3 conductance (green bars in **h** and **j**) relative to that at −40 mV ($n = 9$). Green dots represent individual data and

larger red dots with bars represent the mean ± SEM. **$P < 0.01$, *$P < 0.05$ (repeated measures one-way ANOVA; post-hoc Holm–Šidák test). **k, l** Average conductance relative to the baseline before onset of 5 and 10 mV depolarizing pulses. Black arrowheads show the points at which transient opposing conductance changes were expected to be observed. Shaded areas indicate the mean ± SEM. **m, n** Magnitudes of opposing conductance changes (arrowheads in **k** and **l**) relative to the steady-state conductance (black bars in **k** and **l**) before onsets or offsets of 5 (**m**) or 10 (**n**) mV depolarizing voltage steps ($n = 9$). Green dots represent individual data, and red bars represent the mean ± SEM. **o, p** Average relative current of Kcnh7 channels in response to depolarizing voltage steps. Shaded areas indicate the mean ± SEM. **q, r** Similar to **k** and **l**, but showing the average relative conductance of Kcnh7 currents. Shaded areas indicate the mean ± SEM. **s, t** Magnitudes of opposing conductance changes (arrowheads in **q** and **r**) relative to the preceding steady-state conductance (black bars in **q** and **r**) ($n = 7$). Green dots represent individual data, and red bars represent the mean ± SEM.

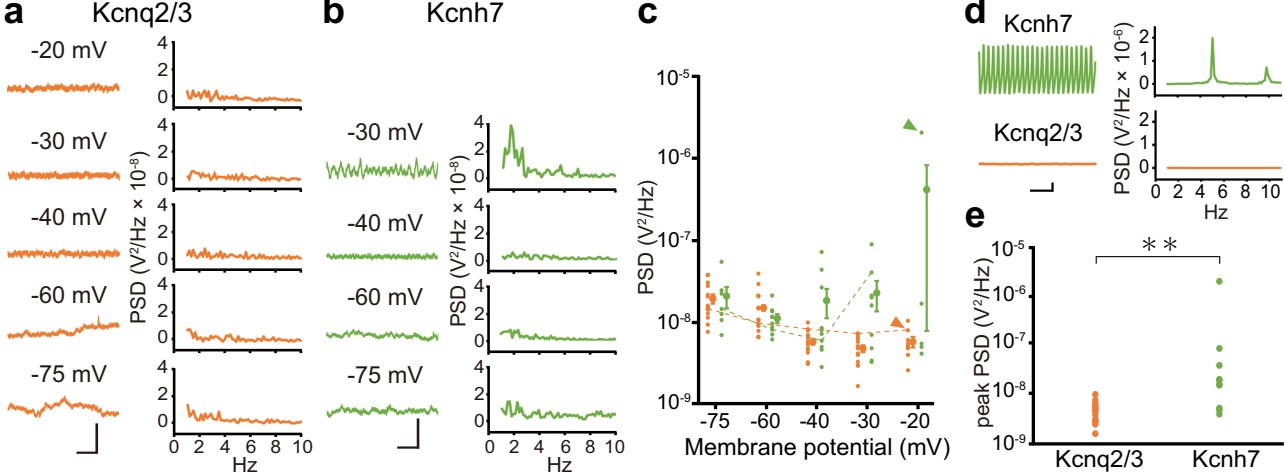

**Fig. 4 | Membrane potential oscillation is not observed in *Kcnq2/3*-expressing HEK293 cells. a** (Left) Representative voltage traces at individual membrane potentials in a *Kcnq2/3*-expressing HEK293 cell. (Right) Power spectrum densities (PSDs) calculated from the left traces by a fast Fourier transform (FFT). **b** Similar to **a**, but with data from a *Kcnh7*-expressing cell. The lack of data at −20 mV was because of unstable current clamp recording at membrane potentials more positive than −20 mV in *Kcnh7*-expressing cells[25]. **c** Magnitude of PSDs at individual membrane potentials in *Kcnq2/3*- and *Kcnh7*-expressing HEK293 cells (*Kcnq2/3*, $n = 8$–$17$, orange; *Kcnh7*, $n = 5$–$10$, green). Small dots represent individual data, and large symbols with bars represent the mean ± SEM. Representative data from **a** and

**b** are connected by orange and green dashed lines, respectively. Orange (*Kcnq2/3*) and green (*Kcnh7*) arrowheads indicate PSDs of representative data in **d**. **d** (Left) Representative traces in *Kcnh7*- (upper, at −20 mV) and *Kcnq2/3*- (lower, at −20 mV) expressing HEK293 cells. (Right) PSDs calculated from the left traces by an FFT. **e** Magnitudes of peak PSDs in each *Kcnq2/3*- and *Kcnh7*-expressing HEK293 cell. The largest peak PSDs among data sampled from −30 mV to −20 mV in individual cells are plotted. Dots represent individual data (*Kcnq2/3*, $n = 15$, orange; *Kcnh7*, $n = 10$, green). **$P < 0.01$ (Mann–Whitney $U$ test). Scale bars, 10 mV, 2 s (**a, b**), 20 mV, 2 s (**d**).

(hyperpolarized) to −20 mV (depolarized) in the current clamp mode. At all membrane potentials, *Kcnq2/3*-expressing cells did not show clear membrane potential oscillations (Fig. 4a, c–e). The membrane potential was more unstable at hyperpolarized potentials than at depolarized ones, resulting in a slight increase in the power spectrum density (Fig. 4a, c). In contrast, not all, but approximately half of *Kcnh7*-expressing HEK293 cells exhibited sustained membrane potential oscillations at depolarized membrane potentials above −30 mV (Fig. 4b–e)[25]. These data suggest that while Kcnq2/3 channels exhibit resonance-like frequency-dependent responses, these are insufficient to generate cell-autonomous membrane potential oscillations.

## Rapid ion channel inactivation and recovery from this inactivation are crucial for the inductor-like gating property and membrane potential oscillation

HEK293 cells express their own endogenous proteins that include ion channels, which may unexpectedly affect resonance and oscillation by binding to and/or co-operating with KCNQ channels. To estimate influences of these factors, we conducted computational simulations

using the NEURON simulator (v8.0)[28]. The previously reported KCNQ model, which incorporates only the gating variable ($m$) for activation (Fig. 5d, Supplementary Table 1)[29], was used for this analysis (Fig. 5a). Sudden current changes at the onset and offset of square voltage steps were observed in a computational simulation (Fig. 5b), consistent with the idea that these changes arise from step voltage changes applied to the activated KCNQ2/3 conductance, as described by Ohm's law. Similar to the experimental analysis (Fig. 3h–l), the model exhibited only slow activation and deactivation without the opposing conductance change in response to voltage steps (Fig. 5c). As reported in previous models including the M-current[21,22,24], the KCNQ model showed the resonance-like frequency-dependent voltage response (Fig. 5f) in response to chirp current injection (Fig. 5e). The voltage response showed a relatively gentle decline after the peak, which was similar to the responses observed in *Kcnq2/3*-expressing HEK293 cells (Fig. 1j, m). Additionally, the KCNQ conductance decreased gradually with increasing input frequency (Fig. 5g). These results indicate that KCNQ channels, even in the absence of an opposing conductance change, can generate resonance-like frequency-dependent responses.

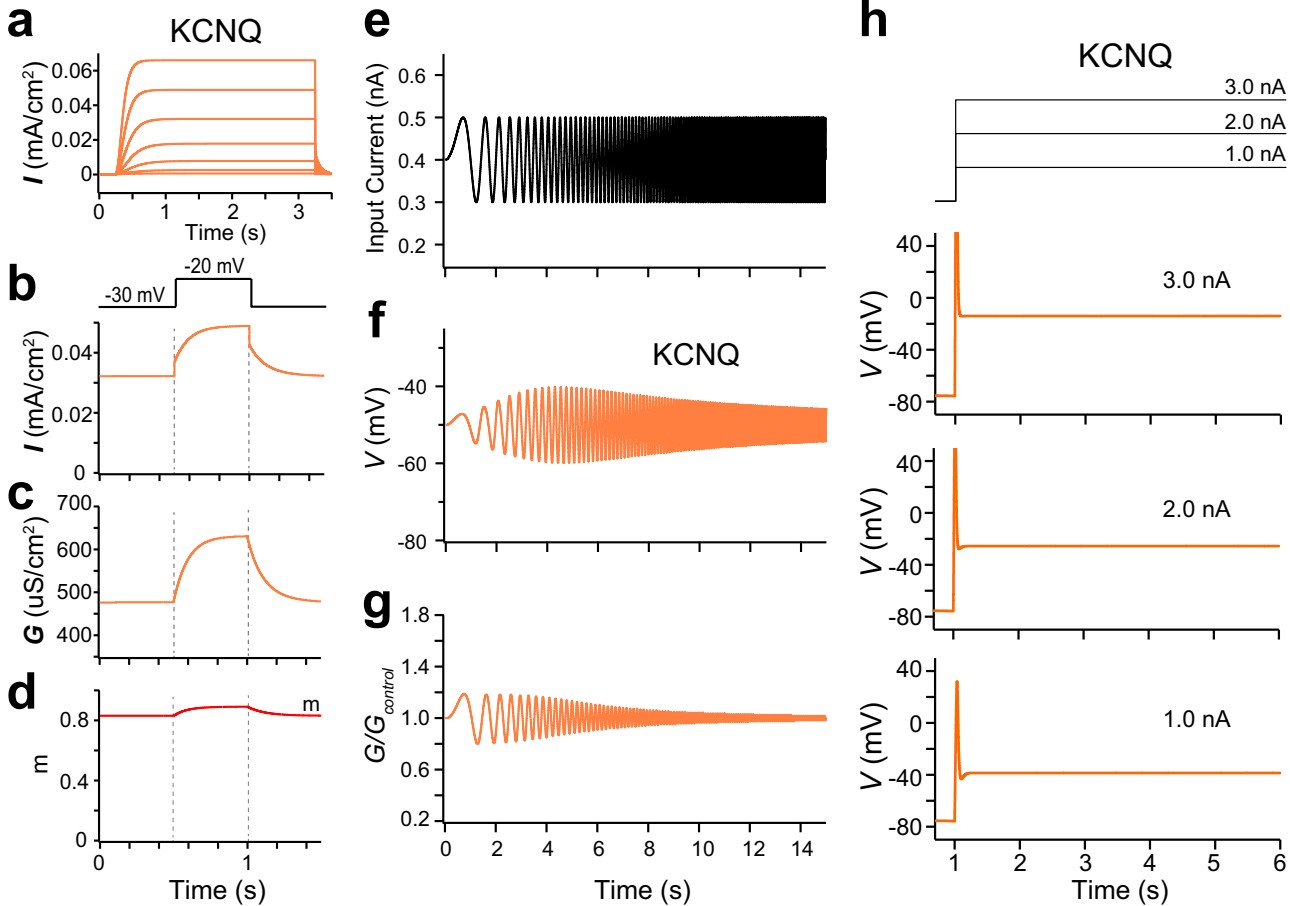

**Fig. 5 | Computational simulation of resonance and oscillation using the KCNQ model. a** Simulated currents using the KCNQ model in response to depolarizing voltage steps of 10 mV for 3 s from −70 to −10 mV by NEURON simulator. Holding potential is −80 mV. **b** Simulated currents using the KCNQ model in response to a square depolarizing voltage step (10 mV) from −30 mV. **c** Conductance changes in response to a depolarizing voltage step from −30 mV in the KCNQ model. **d** Time course of a gating variable (activation and deactivation, *m*) in response to depolarizing voltage steps. **e** Input chirp current. **f** Simulated voltage response to the chirp current input in the KCNQ model. **g** KCNQ conductance relative to the baseline in the KCNQ model. **h** (upper) Step current inputs applied to the KCNQ model. (lower) Simulated membrane potentials for individual current steps.

We next examined the factors underlying the differences in gating behaviors between Kcnq2/3 and Kcnh7 channels using computational analysis. We focused on the property of rapid inactivation and recovery from inactivation, which is inherent to Kcnh7 channels[30–34]. We incorporated a gating variable representing rapid inactivation and recovery (*h*) into the KCNQ model (KCNQ$_{inactivation}$) (Fig. 6d). Unlike the conventional KCNQ channel (Fig. 5a), the current activated by step voltages from −80 mV was strongly suppressed in the KCNQ$_{inactivation}$ model (Fig. 6a). Step current changes at the onset and offset of step depolarization, which were observed in the KCNQ model (Fig. 5b), were substantially diminished (Fig. 6b), instead displaying an exponential rise or decay from the baseline. Furthermore, both the opposing conductance change and subsequent slow activation and deactivation, which were observed in *Kcnh7*-expressing HEK293 cells (Fig. 3q–t), were also observed in this model (Fig. 6c). These data suggest that the unique current response of Kcnh7 channels to square voltage steps is caused by its rapid inactivation and recovery from inactivation.

Consistent with experimental data from *Kcnh7*-expressing HEK293 cells (Fig. 1g, m), the KCNQ$_{inactivation}$ model exhibited the steeper decline in the voltage response at the higher frequency range beyond the resonant frequency than that of the conventional KCNQ model (Figs. 5f and 6f). This response was likely caused by the enhanced conductance of the KCNQ$_{inactivation}$ model around the resonant frequency (Fig. 6g), resulting in finer frequency tuning. Remarkably, the KCNQ$_{inactivation}$ model also exhibited membrane potential oscillation following depolarization from the

resting potential to higher membrane potentials (Fig. 6h). In contrast, similar membrane potential depolarizations failed to elicit oscillation in the conventional KCNQ model (Fig. 5h). These data suggest that the rapid inactivation and recovery dynamics not only enhance frequency tuning but also enable cell-autonomous membrane potential oscillations.

Finally, we experimentally confirmed the crucial role of the rapid inactivation on Kcnh7 channel gating. Rapid inactivation of Kcnh7 channels was reported to be suppressed by bath application of ICA-105574 (2 μM)[35]. As predicted by computational analyses, sudden current changes at the onset and offset of square voltage steps emerged (Supplementary Fig. 2a, b), and opposing conductance changes were suppressed (Supplementary Fig. 2c–f) in *Kcnh7*-expressing cells treated with ICA-105574. These data support the computational analysis conclusion that the rapid inactivation and recovery dynamics are essential for the opposing conductance change of Kcnh7 channels.

## KCNQ2 missense mutations change the membrane potential dependence and frequency preference of resonance

We previously reported that the characteristics of resonance are substantially influenced by differences in the ion channel properties of Kv11 channel subtypes (*Kcnh2* and *Kcnh7*) and splice variants (*Merg1a* and *Merg1b*, two splice variants of *Kcnh2*)[25]. We therefore examined whether the resonance, which was caused by ion channels without the opposing conductance change, might also be modulated by specific ion channel characteristics. To address this, we used LOF and GOF missense *KCNQ2*

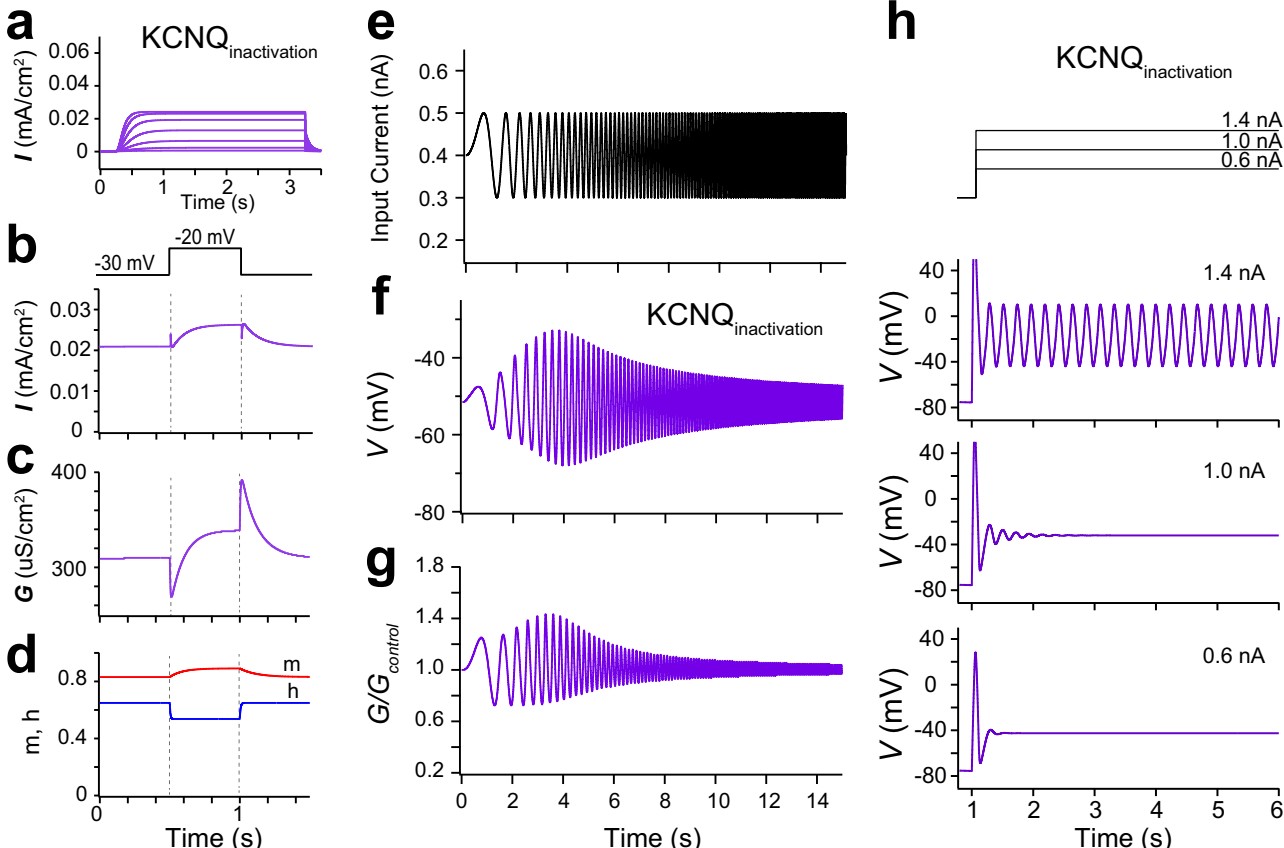

**Fig. 6 | Computational simulation of resonance and oscillation using the KCNQ model with rapid ion channel inactivation. a** Simulated currents using the KCNQ channel with rapid inactivation and recovery (KCNQ$_{inactivation}$) model in response to depolarizing voltage steps same with those in Fig. 5a. **b** Simulated current using the KCNQ$_{inactivation}$ model in response to a square depolarizing voltage step (10 mV) from −30 mV. **c** Conductance changes in response to a depolarizing voltage step from −30 mV in the KCNQ$_{inactivation}$ model. **d** Time courses of gating variables (activation and deactivation, $m$; inactivation, $h$) in response to the depolarizing voltage step. **e** Input chirp current. **f** Simulated voltage response to the chirp current input in the KCNQ$_{inactivation}$ model. **g** Conductance relative to baseline in the KCNQ$_{inactivation}$ model. **h** (upper) Step current inputs applied to the KCNQ$_{inactivation}$ model. (lower) Simulated membrane potentials for individual current steps. Membrane potential oscillations were generated in the KCNQ$_{inactivation}$ model at depolarized membrane potentials.

mutations, which are linked to DEE[9,11–13]. *Human KCNQ2* and *KCNQ3* (*KCNQ2/3*) were co-expressed in HEK293T cells. The R213Q missense mutation in the S4 segment was examined as the LOF mutant, and the R144Q missense mutation in the S2 segment was used as the GOF mutation.

As previously reported, the current-voltage relationship of the R144Q GOF mutant channel was shifted toward more hyperpolarized potentials, whereas that of the R213Q LOF mutant channel was shifted toward more depolarized potentials (Fig. 7a–d, Supplementary Table 2)[9,13]. Consistent with previous reports[13], the activation and deactivation kinetics of the R144Q GOF mutant channel remained unchanged (Fig. 7e, f). In contrast, the R213Q LOF mutant channel exhibited significantly accelerated deactivation kinetics[9,36] (Fig. 7f). In the present study, its activation was also found to be accelerated (Fig. 7e). We next examined how these kinetic differences influenced resonance.

HEK293T cells expressing *human KCNQ2/3* exhibited resonance similar to that observed in HEK293 cells expressing *mouse Kcnq2/3* (Figs. 1i, j and 7i, j). In the R213Q LOF mutant, resonance was generated at more depolarized membrane potentials (Fig. 7k–o), consistent with the depolarizing voltage shift in its activation. In addition, the maximum resonant frequency (14.5 ± 2.0 Hz, $n = 7$) of the R213Q LOF mutant was higher than that of wild-type *KCNQ2/3* (6.5 ± 0.6 Hz, $n = 8$) (Fig. 7p). This higher resonant frequency may be a consequence of the faster deactivation kinetics of the R213Q LOF mutant channel.

The resonant strength of the R144Q GOF mutant tended to be lower than that of wild-type *KCNQ2/3* (Fig. 7m), which may be attributed to reduced viability of R144Q GOF cell cultures. The overactivation of

KCNQ2/3 channels at the resting membrane potential likely compromised cell viability. We therefore primarily focused on the relative membrane potential dependence of resonance (Fig. 7n) rather than the absolute magnitude of resonant strength (Fig. 7m). The R144Q GOF mutant exhibited resonance at more hyperpolarized membrane potentials (Fig. 7g, h, n), although its resonant frequency was comparable to that of the wild-type KCNQ2/3 channel (Fig. 7o, p). These changes were consistent with changes in altered ion channel dynamics induced by the R144Q GOF mutation. Taken together, these data suggest that the resonance caused by ion channels lacking opposing conductance change is also substantially influenced by their specific gating characteristics.

## Discussion
Kcnq2/3 channels showed the band-pass filtering response to chirp current injection, which indicates that they have the potential to produce the resonance-like frequency-dependent response (Figs. 1 and 5). This observation aligns with previous studies suggesting the involvement of the M-current in resonance[14,18–24]. However, our experimental (Figs. 2 and 3) and computational (Fig. 5) analyses indicate that the inductor-like activity of *Kcnq2/3*-expressing HEK293 cells is not completely identical to that of *Kcnh7*-expressing cells.

In electrical circuits, the inductor generates voltage with a polarity that opposes changes in current. The magnitude of this opposing voltage increases linearly with the rate of current change, as described by the equation ($V(t) = -L(dI(t)/dt)$), where $L$ represents the inductance. As a result, rapid current changes at the onset and offset of a square voltage step

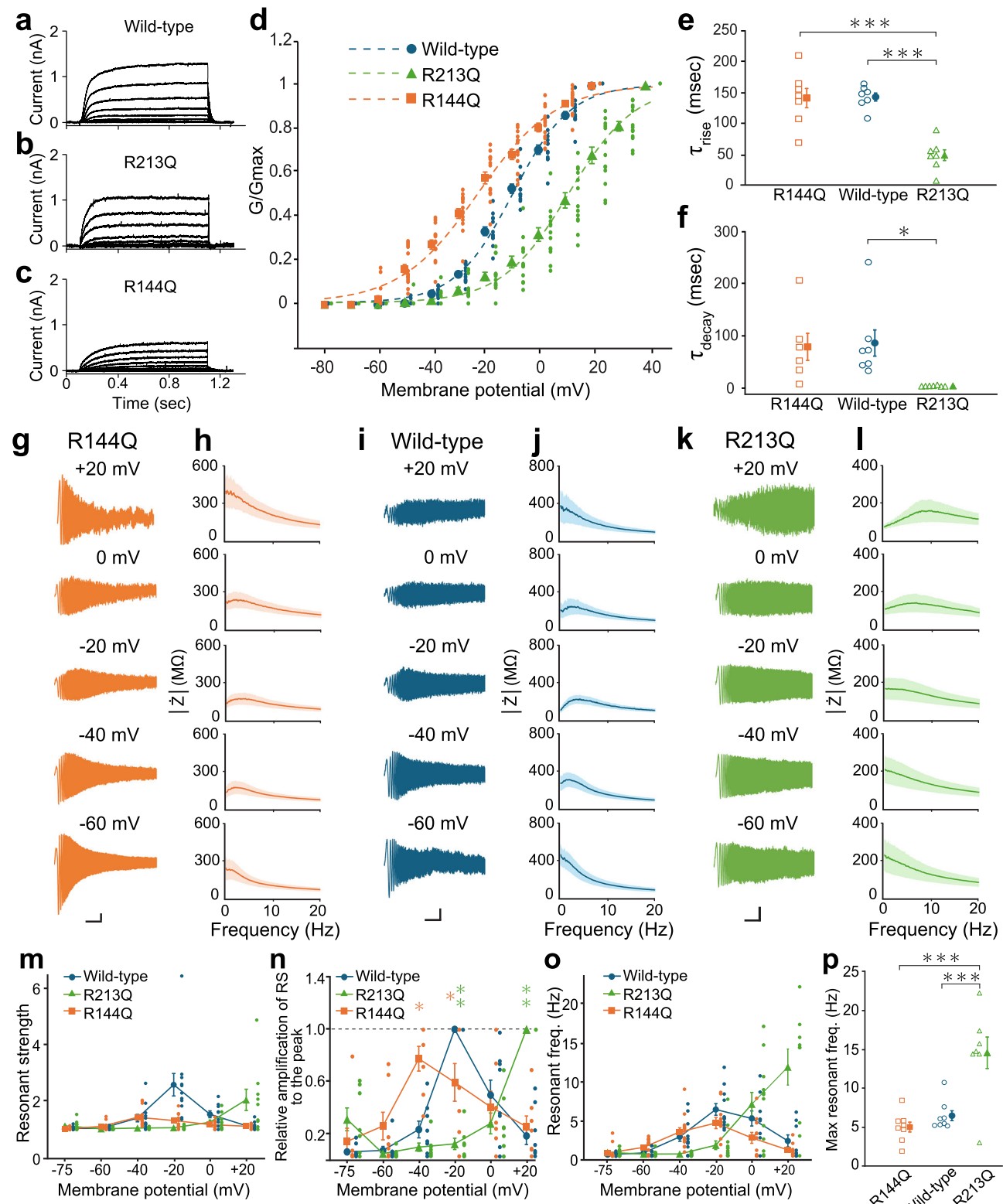

are preferentially suppressed, while steady-state voltage conditions remain unaffected. Meanwhile, the movement of ions flowing through ion channels is passive, and ion channels therefore do not directly generate opposing voltage similar to an inductor. Instead, ion channels that can act as resonating conductance exhibit kinetic performance mimicking that of the inductor. We previously reported that *Kcnh7*-expressing HEK293 cells exhibited 1) the transient conductance change that opposes induced current

changes immediately following the onset of step voltage changes, and 2) the subsequent slow and gradual activation or deactivation toward the steady-state level (Fig. 3q, r)[25]. The fast-opposing conductance change exerts an effect on flowing currents to start the current from the baseline before depolarization onset. The subsequent slow activation or deactivation phase likely mimics recovery from current suppression by the opposing voltage generation by the inductor.

**Fig. 7 | Resonance-like frequency responses for *KCNQ2* loss-of-function (LOF) and gain-of-function (GOF) mutants. a–c** Representative currents evoked in HEK293T cells expressing wild-type (top), R213Q LOF mutant (middle), or R144Q GOF mutant (bottom) *KCNQ2* with wild-type *KCNQ3* in response to the depolarizing voltage protocol (+10 mV steps from –80 mV to +20 mV in wild-type and R144Q, and those from –80 mV to +40 mV in R213Q). **d** The relative conductance-voltage plots of cells expressing wild-type ($n = 12$, blue circles), R144Q ($n = 14$, orange squares), or R213Q ($n = 13$, green triangles) *KCNQ2* with *KCNQ3*. Small dots represent individual data, and symbols with bars represent the mean ± SEM. The superimposed dashed sigmoid curves are theoretical fits obtained using the Boltzmann equation (Supplementary Table 2). **e, f** Rise (**e**) and decay (**f**) time constants estimated by single exponential fitting of currents in R144Q (**e**, $n = 7$; **f**, $n = 6$, orange squares), wild-type ($n = 7$, blue circles), and R213Q ($n = 7$, green triangles) *KCNQ2*-expressing cells. For this analysis, currents were recorded in the high external and low internal $K^+$ condition (see Methods). Empty symbols represent individual cells, and filled symbols represent average values. ***$P < 0.001$, *$P < 0.05$ (one-way ANOVA; post-hoc Holm–Šidák test). **g, i, k** Representative voltage traces in response to chirp currents at individual membrane potentials in R144Q (**g**), wild-type (**i**), or R213Q (**k**) *KCNQ2*-expressing cells. **h, j, l** Averaged Z-F plot of cells expressing R144Q (**h**, $n = 9$–15),

wild-type (**j**, $n = 8$–12), or R213Q (**l**, $n = 9$–13) *KCNQ2* with *KCNQ3*. Shaded areas indicate the mean ± SEM. **m** Membrane potential dependence of the resonant strength in cells expressing wild-type ($n = 8$–12, blue circles), R144Q ($n = 9$–15, orange squares), and R213Q ($n = 9$–13, green triangles) *KCNQ2* with *KCNQ3*. **n** The resonant strength normalized by the largest one in R144Q ($n = 7$, orange squares), wild-type ($n = 8$, blue circles), and R213Q ($n = 7$, green triangles) *KCNQ2*-expressing cells. Values are calculated in individual cells as follows: (resonant strength at individual membrane potentials – 1) / (the largest resonant strength – 1). Each *P*-value indicates the comparison with wild-type KCNQ2 current. **$P < 0.01$, *$P < 0.05$ (Kruskal-Wallis test; post-hoc Holm–Šidák test). **o** Membrane potential dependence of the resonant frequency in cells expressing wild-type ($n = 8$–12, blue circles), R144Q ($n = 9$–15, orange squares), and R213Q ($n = 9$–13, green triangles) *KCNQ2* with *KCNQ3*. In **m–o**, small dots represent individual data, and symbols with bars present the mean ± SEM. **p** Comparison of maximum resonant frequencies. Orange squares, blue circles, and green triangles represent R144Q ($n = 8$), wild-type ($n = 8$), and R213Q ($n = 7$) KCNQ2 currents, respectively. ***$P < 0.001$ (one-way ANOVA; post-hoc Holm–Šidák test). Empty symbols represent individual cells, and filled symbols with bars represent the mean ± SEM. Scale bars, 10 mV, 3 s (**g**). 5 mV, 3 s (**i, k**).

Our experiments indicated that only the latter event occurred in *Kcnq2/3*-expressing cells (Fig. 3d–n). Importantly, our analyses demonstrated that the resonance-like frequency response can be generated in *Kcnq2/3*-expressing cells (Fig. 1i, j) and in the KCNQ computational model (Fig. 5f). These findings suggest that, not the rapid opposing conductance change, but the following slow activation or deactivation is essential to produce the resonance-like frequency-dependent response. How might Kcnq2/3 channels generate the resonance-like frequency-dependent response? Kcnq channels are characterized by slow activation and deactivation kinetics and little inactivation[1,3,37,38], which may be key determinants of their ability to mimic inductor-like performance. The 10%–90% activation and deactivation times of Kcnq2/3 channels in response to the membrane potential depolarization from -80 mV to 0 mV are identical and are approximately 275 ms, which closely corresponds to a quarter cycle of 1.0 Hz oscillation. This temporal alignment enables KCNQ2/3 channels to effectively track sinusoidal membrane potential fluctuations within low frequency ranges (Figs. 2b and 5g). However, at high frequency ranges, the slow kinetics of KCNQ2/3 channels prevent them from keeping pace with the rapid oscillatory changes in membrane potential, resulting in reduced overall channel activation (Figs. 2c–h and 5g). The higher Kcnq2/3 activity may mimic the lower impedance of the inductor at lower frequency ranges, and form the steep upward slope in the Z-F plot (Fig. 1j). At high frequency ranges, the current instead flows through capacitor and forms the downward slope of the Z-F plot.

These findings suggest that ion channels with properties similar to those of Kcnq channels may have the potential to generate Kcnq2/3-type, input frequency-dependent responses. For example, a previous study reported that dendrotoxin-I-sensitive Kv1 voltage-dependent $K^+$ channels exhibited resonance at high frequency ranges (approximately 40 Hz) in mesencephalic V neurons[39]. Kv1 channels are characterized by rapid activation and weak, slow inactivation, making them plausible candidates for mediating Kcnq2/3-type resonance. The higher resonant frequency (41 Hz)[39] than that of Kcnq2/3 channels may be attributed to their faster activation and deactivation kinetics[39–41].

Previous studies have consistently indicated that M-channels (*Kcnq*), Kcnh channels, and HCN channels possess the potential to function as resonating conductance[14,15]. Among these, our findings suggest that the underlying mechanisms by which Kcnq and Kcnh channels contribute to resonance are partially different. Our results indicate that resonating conductance may be functionally subdivided into at least two categories—those mediated by ion channels that exhibit both the opposing conductance change and subsequent slow activation or deactivation (as seen with Kcnh7 channels), and those that exhibit only the slow activation or deactivation (as observed with Kcnq2/3 channels). Meanwhile, the functional classification of HCN channels as resonating conductance remains unresolved. HCN

channels have been widely reported to mediate resonance across numerous brain regions[14,18,19,21,22,42–44]. Among four HCN subtypes, contribution of *HCN1* has been studied using knockout mice[17,45,46]. Voltage responses of HCN-dependent resonance typically exhibit a phase lead relative to chirp current input at the lower frequency range, which suggests that the HCN channel functions as a phenomenological inductor[17,44,47,48]. However, in our previous study, heterologous expression of *HCN1* in HEK293 cells failed to elicit clear membrane potential oscillations or resonance[25]. Consequently, analyses to test the inductor-like gating property (Fig. 3) have not been performed for *HCN1*. It thus remains uncertain whether HCN1 channels emulate Kcnh7-type or Kcnq2/3-type inductor-like performance. Given that HCN channels are subject to extensive modulation by intracellular signaling pathways and auxiliary subunits in neurons[49], it may be possible that certain neuron-specific interacting partners are required for *HCN1* to exhibit inductor-like behavior. Future studies incorporating such factors will be necessary to clarify the precise gating mechanisms underlying *HCN1*-mediated resonance.

Alterations in resonance in *KCNQ2* missense mutations were largely consistent with those predicted from their altered ion channel properties (Fig. 7d, g–p), which suggests that the resonance caused by ion channels lacking the opposing conductance change was also substantially influenced by their gating dynamics. In this study, the R213Q LOF and R144Q GOF mutants shifted the optimal membrane potential for resonance to more depolarized and hyperpolarized levels, respectively. Such membrane potential shifts may alter neural responses to periodic input activities. Notably, the hyperpolarizing shift of resonance induced by the R144Q GOF mutant could enhance the responsiveness of neurons to periodic inputs near the resting membrane potential. This might result in abnormal frequency response of individual cells and contribute to pathological states.

In the present study, the experimental and computational analyses revealed that the opposing conductance change arises from the rapid inactivation and recovery from this inactivation of ion channels (Fig. 6c, d and Supplementary Fig. 2). The inactivation and recovery from inactivation of Kcnh channels occur at very fast rates[31–34]. Continuous activation of Kcnh7 channels at approximately −30 mV might cause weak inactivation, and a small depolarization from −30 mV may quickly cause further Kcnh7 channel inactivation, resulting in a decrease in conductance. These findings suggest that the rapid opposing gating property is a unique feature of ion channel groups with rapid inactivation and recovery dynamics.

Because the fast-opposing conductance change has similar influences on flowing currents to the opposing potential generation of the inductor, we speculated that this property may be a prerequisite for channels to exhibit resonating conductance activity. However, our findings suggest that the opposing conductance change is not essential to produce resonance-like frequency-dependent responses. However, *Kcnh7*-expressing cells had a

steeper downward slope in the Z-F plot (Fig. 1h, m), indicative of finer frequency tuning. This steeper decline in voltage responses was also observed in the KCNQ$_{inactivation}$ model (Fig. 6f). These results indicate that the inductor-like behavior of ion channels substantially contributes to enhancing frequency tuning. Furthermore, the expression of *Kcnh2* variants or *Kcnh7* channels was associated with cell-autonomous membrane potential oscillations at depolarized membrane potentials (Fig. 4b–e)[25]. These oscillations were also observed in the KCNQ$_{inactivation}$ model (Fig. 6h). In contrast, *Kcnq2/3* expression did not lead to regenerative membrane potential oscillation (Fig. 4a, c–e) despite the resonance-like frequency-dependent response. These findings suggest that efficient generation of membrane potential oscillations may require ion channels capable of acting as more ideal inductors.

## Methods

### Cell lines

All experiments were performed in accordance with the guidelines of the biosafety committee for living modified organisms (#2024-229) of Hiroshima University. HEK293 cells (Cat No. JCRB9068, RRID: CVCL_0045) were purchased from JCRB Cell Bank (Osaka, Japan), who stated their authentication. HEK293T cells were provided by Dr. Satoshi Okada (Hiroshima University, Hiroshima, Japan). The cells were cultured in Dulbecco's modified Eagle medium (Thermo Fisher Scientific, Waltham, MA, USA) supplemented with 10% fetal bovine serum, 100 U mL$^{-1}$ penicillin G, and 100 µg mL$^{-1}$ streptomycin under a humidified atmosphere at 37 °C in 5% $CO_2$.

### Transfection

Wild-type *mouse Kcnq2* (NM_001003824.2), *Kcnq3* (NM_152923.3), and *Kcnh7* (NM_133207.2) expression vectors were purchased from Vector-Builder (Chicago, IL, USA). A cytomegalovirus (CMV) promoter was used, and the channel coding sequence was followed by internal ribosome entry sites (IRES) and EGFP (pCMV-*Kcnq2*, pCMV-*Kcnq3*, and pCMV-*Kcnh7*). Wild-type *human KCNQ2* (NM_172107.4) or *KCNQ3* (NM_004519.4) was subcloned into the pCMV-SPORT6 vector harboring bicistronic IRES-driven EGFP or monomeric red fluorescent protein. The *KCNQ2* mutations (c.431 G > A [p.R144Q] or c.638 G > A [p.R213Q]) were introduced by site-directed mutagenesis using a KOD-Plus Mutagenesis Kit (TOYOBO, Osaka, Japan).

HEK293 cells were grown in glass-bottomed dishes (µ-Dish 35 mm low; ibidi, NIPPON Genetics, Tokyo, Japan) for 24 h prior to transfection. They were then transfected with a combination of *Kcnq2* (215 ng per dish) and *Kcnq3* (215 ng per dish) using Lipofectamine LTX (Thermo Fisher Scientific) according to the manufacturer's instructions. *Kcnh7* (430 ng per dish) was transfected alone using the same procedure. For human KCNQ channel expression, HEK293T cells were used, and wild-type, R144Q, or R213Q *KCNQ2* (215 ng per dish) was co-transfected with wild-type *KCNQ3* (215 ng per dish). Whole-cell recordings were obtained 24–72 h after transfection.

### Electrophysiology

Whole-cell recordings were obtained from HEK293 or HEK293T cells using an upright microscope (BX50WI; Olympus, Tokyo, Japan) equipped with an IR-CCD camera system (C2741-60, HAMAMATSU PHOTONICS, Shizuoka, Japan). All recordings were performed at room temperature.

The external solution in all experiments (Figs. 1–4, 7a–d, 7g–p, Supplementary Figs. 1 and 2), except for recordings of the current kinetics through the KCNQ2 mutant channel (Fig. 7e, f), was composed of (in mM) 125 NaCl, 2.5 KCl, 2 CaCl$_2$, 1 MgSO$_4$, 1.25 NaH$_2$PO$_4$, 26 NaHCO$_3$, and 20 glucose and was bubbled with 95% $O_2$ and 5% $CO_2$. For recordings of the current kinetics through the KCNQ2 mutant channel (Fig. 7e, f), the external solution was composed of (in mM) 117.5 NaCl, 10 KCl, 2 CaCl$_2$, 1 MgSO$_4$, 1.25 NaH$_2$PO$_4$, 26 NaHCO$_3$, and 20 glucose. The intracellular solution in all experiments (Figs. 1–4, 7a–d, 7g–p, Supplementary Figs. 1 and 2), except for recordings of the current kinetics through the

KCNQ2 mutant channels (Fig. 7e, f), was composed of (in mM) 115 potassium methansulfonate, 5 KCl, 5 NaCl, 0.5 ethylene glycol-bis-(beta-amino-ethyl ether) N,N,N',N'-tetra-acetic acid (EGTA), 30 4-(2-hydroxyethyl)-1-piperazineethanesulfonic acid (HEPES), 4 MgCl$_2$, 4 2Na-ATP, and 0.4 2Na-GTP (pH 7.3, adjusted with KOH). For recordings of the current kinetics through the KCNQ2 mutant channel (Fig. 7e, f), the intracellular solution was composed of (in mM) 35 potassium methansulfonate, 93 N-methyl-D-glucamine, 5 KCl, 5 NaCl, 0.5 EGTA, 30 HEPES, 4 MgCl$_2$, 4 2Na-ATP, and 0.4 2Na-GTP (pH 7.4, adjusted with HCl). The pipette access resistance was approximately 3–5 MΩ.

Membrane potentials and ionic currents were recorded with an EPC-10 amplifier (HEKA Elektronik, Lambrecht/Pfalz, Germany). The signals were filtered at 3 kHz and digitized at 1 kHz for the resonance experiments (Figs. 1e–m and 7g–p), at 10 kHz for the membrane oscillation experiments (Fig. 4), and at 20 kHz for Kcnq2/3 or Kcnh7 currents in the voltage-clamp experiments (Figs. 1a–c, 2, 3, 7a–f, Supplementary Figs. 1 and 2). On-line data acquisition and off-line data analysis were performed using PATCH-MASTER software (v2×91, HEKA Elektronik). The liquid junction potentials were not adjusted. $G/G_{max}$ plots were fitted using the following Boltzmann equation:

$$\frac{G}{G_{max}} = \frac{1}{1 + exp\left[\left(V_{half} - V_m\right)/k\right]}$$

where $V_{half}$ and $k$ represent the half-conductance potential and slope factor, respectively.

If the electrode potential polarization was >5 mV from the initial value at the end of the recording, the data were omitted from the analysis. Data from recordings with high series resistances (>10 MΩ) were also omitted.

### Impedance measurements

Impedance was measured using ZAP input currents[15,26,27]. A sinusoidal current with a constant amplitude (50 pA) but linearly altered frequency with a range from 0–40 Hz in 40 s (i.e., a chirp current) was applied from a recording electrode under the current clamp mode. Impedance ($\dot{Z}(f)$) was calculated by dividing the fast Fourier transform (FFT) of the voltage response by the FFT of the chirp current. $\dot{Z}(f)$ and $f$ represent the complex impedance and frequency of the input current, respectively. The magnitude of impedance was calculated as follows:

$$\left|\dot{Z}(f)\right| = \sqrt{\left(\dot{Z}(f)_{Re}\right)^2 + \left(\dot{Z}(f)_{Im}\right)^2}$$

where $\dot{Z}(f)_{Re}$ and $\dot{Z}(f)_{Im}$ are the real and imaginary parts of the impedance, respectively.

To minimize artifacts caused by sinusoidal current alterations, another chirp current with a reversed phase was also applied, and the magnitude of impedance was calculated by averaging these two impedances. The frequency (>0.5 Hz) at which impedance reached a maximum was termed the "resonant frequency". The resonant strength was calculated as the ratio of the maximum impedance amplitude at the resonant frequency relative to the impedance amplitude at 0.5 Hz. These calculations were performed using Excel (Microsoft, WA, USA) and Igor Pro 6.37 (WaveMetrics, Portland, OR, USA). Under conditions in which resonance does not occur or is blocked, impedance decreases monotonically, the resonant frequency becomes approximately 0.5 Hz, and resonant strength becomes approximately 1.

### Computational analysis

We performed simulations using the NEURON simulator for Windows (v8.0, https://www.neuron.yale.edu/neuron/)[28]. We used the KCNQ model (KCNQ.mod, ModelDB, #143100) described in Fujita et al.[29] with some modifications of the parameters (Supplementary Table 1). The maximum conductance density of the KCNQ channels ($gmax$) was 0.001 S cm$^{-2}$. The

original KCNQ model only had the gating variable for activation and deactivation ($m$). The KCNQ conductance ($gKCNQ$) was calculated as follows:

$$gKCNQ = gmax * m^4$$

The gating variable $m$ obeys the following kinetics:

$$\frac{dm}{dt} = \frac{m_{\infty}(V) - m}{\tau_m(V)}$$

where $V$ is the membrane potential. The $m_{\infty}$ and $\tau_m$ are the activation or inactivation function and the time constant, respectively, and were calculated using the following equations:

$$m_{\infty}(V) = \frac{1}{1 + exp\left[(\theta_m - V)/k_m\right]}$$

$$\tau_m(V) = \tau_{m0} + \frac{\tau_{m1} - \tau_{m0}}{exp\left[(\varphi_m - V)/\sigma_{m0}\right] + exp\left[(\varphi_m - V)/\sigma_{m1}\right]}$$

To examine the influence of rapid inactivation, a gating variable for inactivation ($h$) was introduced in some experiments ($KCNQ_{inactivation}$, Fig. 6), as follows:

$$gKCNQ_{inactivation} = gmax * m^4 * h$$

$$\frac{dh}{dt} = \frac{h_{\infty}(V) - h}{\tau_h(V)}$$

$$h_{\infty}(V) = h_{min} + \frac{(1 - h_{min})}{1 + exp\left[(\theta_h - V)/k_h\right]}$$

$$\tau_h(V) = \tau_{h0} + \frac{\tau_{h1} - \tau_{h0}}{exp\left[(\varphi_h - V)/\sigma_{h0}\right] + exp\left[(\varphi_h - V)/\sigma_{h1}\right]}$$

To simulate the current-voltage relationship (Figs. 5a and 6a) and the square voltage steps (Figs. 5b–d and 6b–d) in voltage clamp mode, a VClamp point process was used. Ion channels other than KCNQ and KCNQ$_{inactivation}$ were not included to examine the performance of KCNQ channels in response to current and voltage changes. The equilibrium potential of K$^+$ was −97.5 mV and the $dt = 0.01$.

For the simulation of the ZAP stimulus (Figs. 5e–g and 6e–g) and membrane potential oscillations (Figs. 6h and 5h) in current clamp mode, the cell consisted of a cylinder with a 40 μm diameter and 40 μm length, a membrane conductance ($g\_pas$) of 0.000014 S cm$^{-2}$ (corresponding to 947 MΩ, the input resistance of HEK293 cells), a cytoplasmic resistance ($Ra$) of 100 Ω·cm, and a resting membrane potential ($soma\ e\_pas$) of −60 mV. The membrane capacitance of HEK293 cells was approximately 20 pF in the present study, which was too small to clearly observe the declining phase of impedance. Therefore, the membrane capacitance ($cm$) was therefore set to 5 μF cm$^{-2}$. The equilibrium potential of K$^+$ was −97.5 mV, the $v\_init = -60$ mV, and the $dt = 0.01$. The ZAP stimulus (izap.mod, Sinusoidal current implementation in The NEURON Forum, https://www.neuron.yale.edu/ftp/ted/neuron/izap.zip) was injected into the middle of the soma. Membrane potential oscillations in the current clamp mode (Figs. 5h and 6h) were simulated using an IClamp point process.

## Drugs
XE991 (Cat. No. 2000, CAS 122955-13-9) and E-4031 (Cat. No. 1808, CAS 113559-13-0) were obtained from Tocris Bioscience (Bristol, UK). ICA-

105574 (Cat. No. HY-124702, CAS 316146-57-3) was obtained from MedChemExpress (NJ, USA).

## Statistics and reproducibility
All averaged data in the text and figures are presented as the means ± standard error of the mean (SEM). All data in graphs and tables are summarized in Supplementary Data 1. $n$ represents the number of HEK293 or HEK293T cells. The sample sizes and statistical tests were chosen based on previous studies with similar methodologies. Transfection was confirmed by the presence or absence of evoked K$^+$ currents and EGFP expression after whole-cell recordings were established. Cells that exhibited no clear depolarization-evoked currents and were negative for EGFP expression (i.e., non-transfected) in the same dish were used as controls. Recordings were performed from HEK293 or HEK293T cells cultured in three or more dishes. Correlation analysis was performed using the Pearson correlation coefficient. Statistical differences between two groups were assessed using the Mann–Whitney $U$ test, paired $t$-test or Welch's $t$-test, in accordance with the results of Shapiro–Wilk test and the objects that were being compared. Differences among three or more groups in Fig. 7e, f, p and Supplementary Table 2 were assessed using one-way analysis of variance (ANOVA). Differences among three or more groups in Figs. 1k, l and 7n were assessed using Kruskal-Wallis test. Differences among three or more groups in Figs. 2g, h, 3j or 2f were assessed using repeated measures one-way or repeated measures two-way ANOVA, respectively. Significance was set at $P < 0.05$. When a difference among multiple groups was judged to be significant or when a normality test was performed in advance, data were processed using the Holm–Šidák test as a post-hoc test. All tests were two-tailed. Current responses in the RLC circuit were simulated using LTspice XVII (Analog Devices, Norwood, MA, USA). Data analysis was performed using Excel (Microsoft) and Igor Pro 6.37 (WaveMetrics). Statistical analyses were conducted using R (version 4.3.1) and SigmaPlot 12.5 (Systat Software, San Jose, CA, USA).

## Reporting summary
Further information on research design is available in the Nature Portfolio Reporting Summary linked to this article.

## Data availability
All data in graphs and Tables are summarized in Supplementary Data 1. The datasets generated during and/or analyzed during the current study are available from the corresponding author upon reasonable request.

## Code availability
We performed simulations using the NEURON simulator for Windows (v8.0)[28]. We used the KCNQ model (KCNQ.mod, ModelDB, #143100) described in Fujita et al.[29] with some modifications of the parameters (Supplementary Table 1). For simulation of the ZAP stimulus, izap.mod (Sinusoidal current implementation in The NEURON Forum, https://www.neuron.yale.edu/ftp/ted/neuron/izap.zip) was used with some modifications. All codes are available from Zenodo[50].

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

## Acknowledgements

This work was supported by the Japan Agency for Medical Research and Development (AMED) (Grant Number JP24gm6510017 to K.H.). We thank Ms. Miyuki Tsumura for preparing the cell cultures. We also thank Bronwen Gardner, PhD, from Edanz (https://jp.edanz.com/ac) for editing a draft of this manuscript.

## Author contributions

Y.E. contributed to Conceptualization, Data curation, Formal analysis, Investigation, Methodology, Validation, Visualization, Writing—original draft, and Writing—review and editing. Y.K. contributed to Methodology, Resource, Writing—original draft, and Writing—review and editing. S.O. contributed to Methodology, Resource, Supervision, Writing—original draft, and Writing—review and editing. H.M. contributed to Methodology, Resource, Writing—original draft, and Writing—review and editing. K.H. contributed to Conceptualization, Data curation, Formal analysis, Funding acquisition, Methodology, Investigation, Project administration, Supervision, Validation, Visualization, Writing—original draft, and Writing—review and editing.

## Competing interests

The authors declare no competing interests.

## Ethics approval

The research was performed by local researchers throughout the research process.
