## [Transparent Peer Review file · Communications Biology]

***Kcnq* (Kv7) channels exhibit frequency-dependent responses via partial inductor-like gating dynamics**

Corresponding Author: Professor Kouichi Hashimoto

Version 0:

Reviewer comments:

Reviewer #1

(Remarks to the Author)

The manuscript by Eguchi and colleagues describes the resonance properties of *Kcnq2/3* channels mainly by comparison with data on *Kcnh7* channels. Unlike HEK293 cells expressing *Kcnh7* channels, cells containing *Kcnq2/3* show resonance without inductor features. The authors conclude that the two different channel subtypes convey resonance through different mechanism. The authors also investigate mutations of *Kcnq2* channels in the context of neonatal epilepsy. Understanding the individual contributions of ion channel function to network activity and the identification of subtype-specific differences is highly relevant and an important contribution to the neuroscience field, especially in the context of epilepsy. The well-written manuscript conveys a clear message both in text and figures. The experimental design, choice of methods and presented data is of high quality.

A few minor points came to my attention. These should be addressed before publication:

(1) Figure 3 panels O/P and S/T seem to be inconsistent in regards to voltage pulses. I haven't found clear statements about the general inconsistencies with voltage steps used between the two channel types (see also below). Is this due to differences in the activation curve? Please provide more information on adequate voltages and argue for appropriate interpretation.

(2) In figure 4, panel C presents data for *Kcnh7* channel-mediated membrane potential oscillations at a holding potential of -20 mV, while presented *Kcnq2/3* signals were recorded at -30 mV. This inconsistency should be explained, especially since oscillations at -30 mV is also much lower according to Matsuoka et al. (2021, DOI: 10.1113/JP280342). In addition, the shown example for *Kcnh7* appears to be something like an "outlier" based on data points presented in panel D. This presentation makes it difficult to appreciate the differences. Data for *Kcnh7* channels should, if possible, be added to panel B as well.

(3) Figure 5 title states "Simulation of KCNQ currents with or without rapid ion channel inactivation". I don't see data or explanations about the presence or absence of rapid ion activation in this figure. Maybe this is more related to a comparison between figs 5 and 6?

Reviewer #2

(Remarks to the Author)

I found the work by Eguchi et. al. a convincing experimental exploration of the resonant properties of KCNQ channels, revealing a clear relationship between the dynamics of these channels and a cell's frequency-dependent response. The experimental results are clear and convincing, and I will leave any precise critiques of the methodology to my fellow reviewers (as my expertise is primarily computational).

Despite this, I have two main elements of serious concern about the manuscript. The first is general: I struggled to discern the mechanistic advance or open question addressed in the manuscript, instead finding it to be largely "descriptive" in nature. This is, potentially, an issue of presentation (and therefore could be addressed in revision), but it is significant nonetheless, especially because it prevented me from affirmatively answering important questions regarding the community's interest in these results and how they might advance the field. I outline specific concerns below:

1) The paper's argument revolves strongly around the relationship between the m-channel and "inductor-like" activity. This terminology is presented without any introduction or explanation, making the manuscript very difficult to decipher for someone not already familiar with this terminology, which I do not believe to be ubiquitous. While some effort is made to address this in the discussion, this choice is potentially problematic for the general audience of Communications Biology, rather than a journal geared primarily towards biomedical engineers.

2) Building off of the above, the main finding of the paper as I read it was that KCNQ channels do not necessarily act like inductors, in contrast to KCNH channels. Contextualizing the paper in this way is problematic on two fronts, in my opinion. First, the choice to compare to KCNH channels seems somewhat arbitrary, and based simply on the fact that this was a channel previously characterized by the authors. While I would bet that there is a further justification for this choice, it does not come through in the text as currently written. Second, there are a wide range of ion channels that can contribute to subthreshold resonance as characterized in this manuscript: as just one example, recent experimental and computational work has rigorously probed how the h-channel contributes to this dynamic, and how differences between the human and rodent settings can be attributed to the kinetics of this channel (Kalmbach et al., 2018; Moradi Chameh et al., 2021; Rich et al., 2021). The claim that the lack of inductor-like behavior in KCNQ channels is unusual and/or a significant advance in our understanding of the mechanisms underlying subthreshold resonance relies on the assumption that inductor-like behavior is the norm underlying subthreshold resonance, but minimal evidence to this point is presented beyond the somewhat arbitrary example of KCNH channels. Given the extensive literature on h-channels, would they be classified as inductors? What about other mechanisms yielding subthreshold resonance? This additional context would be necessary to draw the broader conclusions alluded to in this study; otherwise, it is simply a comparison between the mechanism of resonance in two specific channels.

3) Many of the conclusions drawn from the results seem to be classic examples of "overselling". As perhaps the most relevant example, there is no one-to-one relationship between the subthreshold resonance studied here and suprathreshold resonance that affects spiking activity (Rotstein 2017; Rotstein and Nadim 2014)... related to the citations above, one could compare the subthreshold findings in Rich et al., 2021 to the follow-up study on suprathreshold frequency-preference in Inibhunu et al. 2023. Thus, the contextualization of the impact of these results relative to epilepsy is tenuous, at best. While m-channelopathies undoubtedly are relevant in epilepsy, this effect is typically related to the effects of the m-current on suprathreshold spike-frequency adaptation (Lerche et al. 2001, Rogawski et al. 2000, Steinlein et al. 2000, Byers et al. 2021) and not subthreshold resonance. If this is the argument the authors wish to make to address questions regarding these results "influence on the field," further work is needed to justify this argument based on the literature.

My second serious concern regards the computational results (my area of expertise). Broadly, I struggled to see what the computational results added to the manuscript: as presented, they largely serve to replicate the experimental findings. While this no doubt has some value, it is minimal, and certainly does not advance the argument of the paper. There were also significant issues with how these results were presented that eschew accepted conventions in the field which I found troubling. Specifically:

1) Referring to the computational results as "theoretical" is at worst incorrect, and at best misleading. In the field, "theoretical" results almost universally refer to conclusions drawn from mathematical analyses of hypothetical, abstract systems. This is not what is done in this work: this work should be presented as "computational" or "in silico" results, reflecting that they are investigations of the outputs of a model neuron.

2) The lack of detail on the computational model used and its implementation was very conspicuous. With what level of spatial precision was the cylindrical model simulated (i.e., what how many "components" were there in the multi-compartment model, if any)? What justifies the choice of cylindrical shape used? Perhaps most importantly, what other ionic channels are included in this model cell? These details not only are necessary for the replication of this work (which, by the way, should be facilitated by storing the related code in an openly-accessible setting, as is convention), but would certainly influence the results of these simulations.

2a) Another important related example: In Figure 3, it is mentioned that "currents were simulated using LTSpice XVII", without any further detail. This is entirely insufficient to describe a computational experiment. The entirety of the details of the simulation and the simulation environment (case and point, this is the first I've heard of LTSpice) should be included so that they can be replicated.

3) As presented, the computational model is used largely to replicate findings already observed experimentally. This does nothing to advance any sort of argument of the paper, and is a very superficial use of computational modeling. While the alteration of the KCNQ channel into an inactivating channel is interesting, the presented results are once again entirely superficial. The computational setting allows for a wealth of additional explorations that would, in fact, be very useful in this manuscript in deciphering the actual mechanism at play relating m-channel activity to resonance, but there is no effort made to utilize these resources.

As the manuscript currently stands, these two criticisms are, unfortunately, quite serious. When there are no underlying scientific issues, but rather issues with presentation and/or argumentation, my philosophy is usually to defer to the author and provide an opportunity to address these concerns in a revision. I believe that a thorough revision could certainly address my first major concern in order to clearly delineate the hypothesis motivating this work and how it advances our understanding of resonance in neurons, rather than the current version that is overly descriptive. However, I do have significant worries about how the computational component of the paper integrates with the experimental component: in

order for the computational element to serve a vital role in the paper's argument rather than simply feel "tacked on", more computational experiments would certainly be needed that probe the relationship between the m-current and resonance in a manner that complements the experimental argument, rather than merely replicating it. As such, I imagine that the final decision on this manuscript will come down to the opinions of my fellow reviewers regarding the experimental component of the paper, which is the majority and therefore should be weighed more significantly than my concerns about the computational element.

While I'm disappointed I cannot be more positive in my review at this time, I want to applaud the authors for what was clearly a significant undertaking, as well as for studying what I believe to be a very interesting (and perhaps understudied) question in the field. It's my sincere hope that these comments will lead to an improved manuscript regardless of its final home.

Reviewer #3

(Remarks to the Author)

The paper discuss the ability of specific ion channels to create resonance, only have some frequencies pass the membrane, the exact mechanism in discussed.

The paper and the results, are interesting, and well conducted.

I will suggest to expand the NEURON model a bit more, the experimental results showed many possible mechanisms and some were suggested as responsible. And different KO experiments. The model can be extended to show replication and alignment of the theories.

The modeling work required is not intense, as this is a single neuron, I suggest to create 1/2 notebook in which you replicate more experiments and show the results.

Currently no analysis/model code is supplied it should extended be part of the paper.

Thank you.

Version 1:

Reviewer comments:

Reviewer #1

(Remarks to the Author)

I thank the authors for carefully implementing the feedback for better clarity of the presentation of the data. All the issues raised by me are now resolved and I consider the article suitable for publication.

Reviewer #2

(Remarks to the Author)

I applaud the authors for the effort and care that they've taken in the revision process, yielding a significantly improved manuscript. In particular, I found the new text more precisely defining the "inductor-like" terminology to be of great use to the interpretability of the paper's overall argument, and the contextualization of the computational results is also much more apparent in the revised manuscript.

There are, however, some serious concerns that I raised in my initial review that the authors did not address in their revision. While these decisions were explained in the "rebuttal" document, I unfortunately found such explanations insufficient for the three specific, important points I outline below. It is my opinion that these points need to be fully addressed in the manuscript not only for scientific rigor, but for it to meet the stated standards for publication in *Communications Biology*, particularly regarding this work's ability to influence thinking in the larger field of neurophysiology.

1) In order for this manuscript to be broadly impactful, rather than just of interest to those who study KCNQ and KCNH channels, the findings must be properly contextualized relative to the entire literature relating to subthreshold resonance. The authors state in their rebuttal that, "It has been assumed that the resonant property of the cell membrane is generated by ion channels acting as inductors" without any citations or justification. I, in fact, would strongly disagree with this assertion, and posit that there are multiple proposed mechanisms by which subthreshold resonance can arise in neurons through the activity of differing ion channels. At bare minimum, statements of this type need to be more fully justified in the manuscript, and my strong suggestion is that the authors fully describe the multiple ways in which this dynamic can arise in neurons (including the T-type channel, which they mention in the manuscript, as well as the h-channel, which I raised in my previous review with citations) and how their findings fit in this larger literature.

2) The statements made by the authors in their rebuttal document regarding the relationship between the h-channel and subthreshold resonance are in my view inaccurate. The work I mentioned in my initial review includes analysis showing that the dynamics of the h-channel indeed are sufficient to drive subthreshold resonance in itself. In fact, the computational strategy taken by Kalmbach et. al. to show this closely parallels the strategy the authors took to make a similar assertion for KCNQ channels in this manuscript. If the authors wish to (reasonably, in my opinion) use their computational results to show that KCNQ channels alone are sufficient to produce subthreshold resonance, they cannot also disregard similar results from Kalmbach et. al. and the related studies I initially mentioned. The reality that the h-channel can induce subthreshold resonance, and how this does or does not support the authors assertions regarding the relationship between inductor-like activity and subthreshold resonance, is something that I believe must be addressed in the Discussion of the manuscript to

fully and properly contextualize these results relative to contemporary literature, particularly given the similarities in the computational strategies that I describe above.

3) The revision to the text the authors made in response to point 3 in my original review misses the point. The new text references "the responsiveness of neurons to periodic inputs near the resting membrane potential, which might lead to abnormal increases in synchronized network activity." However, the literature, particularly the citations I referenced previously, do not support this statement: there is not necessarily a one-to-one relationship between the sub- and supra-threshold frequency preferences of neurons, particularly as characterized by the response to a ZAP input. As another example, the statement on Lines 50-52 that, "Resonance is thought to be crucial for synchronized activity of multiple neurons across networks and for generating cell-autonomous oscillations at characteristic frequencies" is not inherently true given the non uniform relationship between sub- and supra-threshold resonance. Properly contextualizing their results relative to this distinction is necessary for this work to be of interest to the broader field. While my opinion is that the results presented here are insufficient to make any claims about synchronized network activity at the level of neuronal spikes, please note this isn't a criticism of the quality or impact of the work at all; rather, as initially mentioned, this is simply a caution to not unnecessarily "oversell" the results. I would strongly encourage the authors review the citations I included in my original review relative to this point and adjust their presentation of their results accordingly.

While I feel strongly about the necessity for these three points to be fully addressed, I also acknowledge that my assessment is somewhat of an outlier relative to the other two peer reviewers. Thus, I am comfortable letting the editor decide how significant the revisions to address the above points need to be and whether they can be made as a "minor revision" or whether another round of peer review is necessary. If it is the later, I would ask the authors include a "tracked changes" version of the manuscript with their resubmission, as is typically required, to highlight new and revised text in the manuscript.

Reviewer #1 (Remarks to the Author):

The manuscript by Eguchi and colleagues describes the resonance properties of Kcnq2/3 channels mainly by comparison with data on Kcnh7 channels. Unlike HEK293 cells expressing Kcnh7 channels, cells containing Kcnq2/3 show resonance without inductor features. The authors conclude that the two different channel subtypes convey resonance through different mechanism. The authors also investigate mutations of Kcnq2 channels in the context of neonatal epilepsy. Understanding the individual contributions of ion channel function to network activity and the identification of subtype-specific differences is highly relevant and an important contribution to the neuroscience field, especially in the context of epilepsy.

The well-written manuscript conveys a clear message both in text and figures. The experimental design, choice of methods and presented data is of high quality.

A few minor points came to my attention. These should be addressed before publication:

(1) Fig. 3 panels O/P and S/T seem to be inconsistent in regards to voltage pulses. I haven't found clear statements about the general inconsistencies with voltage steps used between the two channel types (see also below). Is this due to differences in the activation curve? Please provide more information on adequate voltages and argue for appropriate interpretation.

[Our response to the comment]

We appreciate this helpful suggestion and acknowledge the typographical error in the initial holding voltage for the Kcnh7 experiments shown in Fig. 3s/t. The correct value, as indicated in Fig. 7o/p, is “-30 mV.” We have now corrected these in Fig. 3. In these experiments, voltage pulses were initiated from -40 mV for Kcnq2/3 and -30 mV for Kcnh7, which correspond to membrane potentials at which HEK293 cells expressing Kcnq2/3 or Kcnh7 (Fig. 11) (Matsuoka *et al.*, 2021) exhibited the largest resonant strength.

To clarify this rationale, we have now explicitly explained selection of initial holding voltages in the text (page 8, line 175–177, 179–181) as follows:

“Square voltage steps of ± 5 or ± 10 mV were applied from a holding potential of -40 mV (Fig. 3d, e), the condition where Kcnq2/3-expressing cells exhibited the largest resonant strength (Fig. 11).”

“For comparison, we conducted parallel experiments using HEK293 cells expressing Kcnh7 at a holding potential of -30 mV, where Kcnh7-expressing cells exhibited the

largest resonant strength (Fig. 11)²³.”

(2) In fig. 4, panel C presents data for *Kcnh7* channel-mediated membrane potential oscillations at a holding potential of -20 mV, while presented *Kcnq2/3* signals were recorded at -30 mV. This inconsistency should be explained, especially since oscillations at -30 mV is also much lower according to Matsuoka et al. (2021, DOI: 10.1113/JP280342). In addition, the shown example for *Kcnh7* appears to be something like an “outlier” based on data points presented in panel D. This presentation makes it difficult to appreciate the differences. Data for *Kcnh7* channels should, if possible, be added to panel B as well.

[Our response to the comment]

We have revised Fig. 4 in accordance with the reviewer’s suggestion. We added representative data of a *Kcnh7*-expressing cell recorded at individual membrane potentials (Fig. 4b). These representative data are connected by dotted lines in Fig. 4c. In these representative data recorded from a *Kcnh7*-expressing cell, data at -20 mV is missing because voltage recordings at membrane potentials more positive than -20 mV were often unstable in *Kcnh7*-expressing cells, making current-clamp recordings difficult (Matsuoka *et al.*, 2021).

Additionally, the representative data of *Kcnq2/3*-expressing cells in Fig. 4c was replaced with data sampled at the membrane potential of -20 mV. Unlike *Kcnh7*-expressing cells, recordings from *Kcnq2/3*-expressing cells were stable at such strongly depolarized membrane potentials. These representative data shown in Fig. 4d are indicated by arrowheads in Fig. 4c. Because the ordinate of Fig. 4d is presented on a logarithmic scale, *Kcnh7* exhibits significantly larger PSDs compared to *Kcnq2/3*, even near the threshold. These data suggest that not all, but about a half of *Kcnh7*-expressing cells exhibited membrane potential oscillation, which is consistent with our previous study (Matsuoka *et al.*, 2021).

(3) Figure 5 title states “Simulation of KCNQ currents with or without rapid ion channel inactivation”. I don’t see data or explanations about the presence or absence of rapid ion activation in this figure. Maybe this is more related to a comparison between figs 5 and 6?

[Our response to the comment]

Thank you for pointing out these important issues. The titles of Fig. 5 and 6 were incorrect, and we have now revised them as follows:

Fig. 5 Title: Computational simulation of resonance and oscillation using the KCNQ model

Fig. 6 Title: Computational simulation of resonance and oscillation using the KCNQ model with rapid ion channel inactivation

Reviewer #2 (Remarks to the Author):

I found the work by Eguchi et. al. a convincing experimental exploration of the resonant properties of KCNQ channels, revealing a clear relationship between the dynamics of these channels and a cell's frequency-dependent response. The experimental results are clear and convincing, and I will leave any precise critiques of the methodology to my fellow reviewers (as my expertise is primarily computational).

Despite this, I have two main elements of serious concern about the manuscript. The first is general: I struggled to discern the mechanistic advance or open question addressed in the manuscript, instead finding it to be largely "descriptive" in nature. This is, potentially, an issue of presentation (and therefore could be addressed in revision), but it is significant nonetheless, especially because it prevented me from affirmatively answering important questions regarding the community's interest in these results and how they might advance the field. I outline specific concerns below:

1) The paper's argument revolves strongly around the relationship between the m-channel and "inductor-like" activity. This terminology is presented without any introduction or explanation, making the manuscript very difficult to decipher for someone not already familiar with this terminology, which I do not believe to be ubiquitous. While some effort is made to address this in the discussion, this choice is potentially problematic for the general audience of Communications Biology, rather than a journal geared primarily towards biomedical engineers.

[Our response to the comment]

The "M-current", mediated by KCNQ voltage-dependent K^+ channels, was originally identified by Brown and Adams, who discovered that it is suppressed by the activation of muscarinic receptors (Brown & Adams, 1980). We have updated the description of the M-current in the Introduction section (page 3, line 32–34) as follows:

"The activation of various G protein-coupled receptors, including muscarinic receptors, depolarizes certain neurons by suppressing the activity of a low-threshold, non-inactivating K^+ conductance called 'M-current'¹"

In live cells, resonance has been proposed to emerge from biophysical properties of the plasma membrane, which can be modeled as a parallel resonant circuit composing a resistor, inductor, and capacitor (RLC circuit), as illustrated in Fig. 3a. The plasma membrane is biophysically modeled as an electrical circuit with a resistor and capacitor

in parallel. While live cells do not possess a true inductor, certain ion channels are believed to function as the inductive component, mimicking “inductor-like behavior” (Hutcheon & Yarom, 2000). We also revised descriptions about this inductive component in the Introduction section. (page 3, line 52 – page 4, line 57) as follows:

“In neurons, resonance has been proposed to arise from the biophysical properties of the plasma membrane, which is conceptually equivalent to a parallel resonant circuit consisting of a resistor, inductor, and capacitor (RLC circuit)¹⁴. The plasma membrane is biophysically represented as an electrical circuit with a resistor and capacitor in parallel. In living cells, the inductive component is proposed to be mediated by ion channels capable of mimicking inductor-like behavior.”

2) Building off of the above, the main finding of the paper as I read it was that KCNQ channels do not necessarily act like inductors, in contrast to KCNH channels. Contextualizing the paper in this way is problematic on two fronts, in my opinion. First, the choice to compare to KCNH channels seems somewhat arbitrary, and based simply on the fact that this was a channel previously characterized by the authors. While I would bet that there is a further justification for this choice, it does not come through in the text as currently written. Second, there are a wide range of ion channels that can contribute to subthreshold resonance as characterized in this manuscript: as just one example, recent experimental and computational work has rigorously probed how the h-channel contributes to this dynamic, and how differences between the human and rodent settings can be attributed to the kinetics of this channel (Kalmbach et. al., 2018; Moradi Chameh et. al., 2021; Rich et al., 2021). The claim that the lack of inductor-like behavior in KCNQ channels is unusual and/or a significant advance in our understanding of the mechanisms underlying subthreshold resonance relies on the assumption that inductor-like behavior is the norm underlying subthreshold resonance, but minimal evidence to this point is presented beyond the somewhat arbitrary example of KCNH channels. Given the extensive literature on h-channels, would they be classified as inductors? What about other mechanisms yielding subthreshold resonance? This additional context would be necessary to draw the broader conclusions alluded to in this study; otherwise, it is simply a comparison between the mechanism of resonance in two specific channels.

[Our response to the comment]

It has been assumed that the resonant property of the cell membrane is generated by ion channels acting as inductors. However, due to the lack of detailed electrophysiological

analyses on individual ion channels, this assumption has remained unverified. Our study directly challenges this view by demonstrating that resonance-like frequency responses are not universally governed by a single mechanism. Instead, the key ion channel properties underlying resonance vary across different ion channels.

To date, only a limited number of ion channels have been consistently reported as resonating conductance, including M-channels (*Kcnq*), *Kcnh* channels, hyperpolarization-activated cyclic nucleotide-gated potassium (*HCN*) channels and T-type voltage dependent Ca^{2+} channels (VDCCs) (Hutcheon & Yarom, 2000; Hashimoto, 2019). Regarding T-type VDCCs, we previously demonstrated that they function as amplifying conductance, enhancing but not directly generating resonance (Matsumoto-Makidono *et al.*, 2016). Among these candidate ion channels, *Kcnh7* is the only one that has been experimentally proven to exhibit inductor-like gating properties at the depolarized membrane potential range (rather than subthreshold levels) (Matsuoka *et al.*, 2021). This was a key reason for selecting *Kcnh7* as a representative ion channel for comparison.

As the reviewer pointed out, *HCN* channels (h-channel) are well-established as a major resonating conductance across various brain regions (Hutcheon & Yarom, 2000; Hashimoto, 2019). However, we did not include *HCN* channels as representative inductor-like channels in our heterologous expression experiments for the following reasons. In our previous studies on inferior olivary neurons, we found that resonance at hyperpolarized membrane potentials was abolished by ZD7288, an HCN-specific blocker, and was impaired in *HCN1* knockout mice, confirming a critical role for HCN channels in mediating resonance (Matsumoto-Makidono *et al.*, 2016). However, when we expressed *HCN1* in HEK293 cells—the same system used in the present study—we observed no clear resonance in the majority of cells (Matsuoka *et al.*, 2021). Membrane potential oscillation was also not observed. These data indicate that *HCN1* expression alone is insufficient to induce resonance and oscillation. We now speculate that *HCN* channels may require auxiliary factors to exhibit inductor-like gating property, but these factors remain unidentified. Due to this limitation, we could not use *HCN* channels as a representative example of ion channels with inductor-like activity in our heterologous expression experiments.

In this study, we analyzed the *Kcnq2/3* channel, whose roles in resonance have not been well understood, by comparing them with the function of *Kcnh7*, whose inductor-like function has been clearly identified. Therefore, we believe that comparisons across ion channels were conducted using the most appropriate experimental models, given the current state of knowledge. The explanation of why *Kcnh7* is used is revised

as follows (page 5, lines 86-88):

*“For comparison, in some experiments, we introduced *Kcnh7* into HEK293 cells, which has been experimentally reported to exhibit inductor-like gating properties and generate resonance and membrane potential oscillations²³.”*

3) Many of the conclusions drawn from the results seem to be classic examples of "overselling". As perhaps the most relevant example, there is no one-to-one relationship between the subthreshold resonance studied here and suprathreshold resonance that affects spiking activity (Rotstein 2017; Rotstein and Nadim 2014)... related to the citations above, one could compare the subthreshold findings in Rich et al., 2021 to the follow-up study on suprathreshold frequency-preference in Inibhunu et. al. 2023. Thus, the contextualization of the impact of these results relative to epilepsy is tenuous, at best. While m-channelopathies undoubtedly are relevant in epilepsy, this effect is typically related to the effects of the m-current on suprathreshold spike-frequency adaptation (Lerche et. al. 2001, Rogawski et. al. 2000, Steinlein et. al. 2000, Byers et. al. 2021) and not subthreshold resonance. If this is the argument the authors wish to make to address questions regarding these results "influence on the field," further work is needed to justify this argument based on the literature.

[Our response to the comment]

Individual *KCNQ2* mutations associated with self-limited neonatal epilepsy (SLNE) and developmental and epileptic encephalopathy (DEE) have been reported to exert diverse effects on channel activity (Miceli *et al.*, 2013; Abidi *et al.*, 2015; Miceli *et al.*, 2022). Some mutations lead to loss-of-function (LOF) effects by inducing a depolarizing shift in the current-voltage relationship, thereby reducing *KCNQ* activity near the threshold. These LOF mutations can readily explain epileptic phenotypes, as they suppress peri- to supra-threshold *KCNQ* activity, leading to reduce spike firing adaptation and ultimately enhanced neuronal excitability, as noted by the reviewer.

Importantly, some other *KCNQ2* mutations exhibit gain-of-function (GOF) effects, enhancing subthreshold *KCNQ* activity by shifting the current-voltage relationship in a hyperpolarizing direction. The epileptogenic mechanism of these GOF mutations remains an open question, as they would be expected to suppress spike firing near threshold, which contradicts the typical basis of epileptic activity. Several hypotheses have been proposed to resolve this paradox (Miceli *et al.*, 2015; Varghese *et al.*, 2023), but the underlying mechanism remains unresolved. In this study, we explore

the possibility that GOF mutations enhance peri- to subthreshold resonance, thereby facilitating synchronized neuronal activation. Such enhanced synchronization could contribute to epileptic activity, even if the overall spike firing frequency is reduced by GOF mutation.

However, it is important to emphasize that investigating epilepsy mechanisms was not the primary objective of this study. In our previous work, we demonstrated that resonance mediated by ion channels with inductor-like gating properties is strongly influenced by the intrinsic channel properties (Matsuoka *et al.*, 2021). This point was assessed by comparing the properties of different *Kcnh* subtypes (*Kcnh1a*, *Kcnh1b* and *Kcnh7*) in the previous study. To extend this concept to *KCNQ2/3*, which partially lack inductor-like gating properties, we examined representative LOF and GOF *KCNQ2* mutations. This approach is more appropriate to address this issue than that conducted in our previous study, which used other subtypes. Our findings successfully demonstrated that resonance mediated by *KCNQ2/3*—despite lacking inductor-like gating properties—was also strongly influenced by the intrinsic characteristics of ion channels (Fig. 7).

To clarify the scope of our investigation, we have now withdrawn the subheading and most of discussions related to epilepsy (page 16, line 357–362).

“Notably, the hyperpolarizing shift of resonance induced by the R144Q GOF mutant could enhance the responsiveness of neurons to periodic inputs near the resting membrane potential, which might lead to abnormal increases in synchronized network activity. This finding may highlight the potential role of KCNQ2 GOF mutations in modulating neural network dynamics and contributing to pathological states.”

My second serious concern regards the computational results (my area of expertise). Broadly, I struggled to see what the computational results added to the manuscript: as presented, they largely serve to replicate the experimental findings. While this no doubt has some value, it is minimal, and certainly does not advance the argument of the paper. There were also significant issues with how these results were presented that eschew accepted conventions in the field which I found troubling. Specifically:

- 1) Referring to the computational results as "theoretical" is at worst incorrect, and at best misleading. In the field, "theoretical" results almost universally refer to conclusions

drawn from mathematical analyses of hypothetical, abstract systems. This is not what is done in this work: this work should be presented as "computational" or "in silico" results, reflecting that they are investigations of the outputs of a model neuron.

[Our response to the comment]

Thank you for these very helpful suggestions. We now replace all “theoretical” to “computational” or “in silico” in the revised manuscript.

2) The lack of detail on the computational model used and its implementation was very conspicuous. With what level of spatial precision was the cylindrical model simulated (i.e., what how many "components" were there in the multi-compartment model, if any)? What justifies the choice of cylindrical shape used? Perhaps most importantly, what other ionic channels are included in this model cell? These details not only are necessary for the replication of this work (which, by the way, should be facilitated by storing the related code in an openly-accessible setting, as is convention), but would certainly influence the results of these simulations.

[Our response to the comment]

We used NEURON simulator for computational analyses in this study (Carnevale & Hines, 2006). While the NEURON simulator provides a graphical user interface for parameter adjustments, we primarily utilized .hoc and .mod files to define our model. One limitation of the NEURON simulator is that it only allows a cylindrical soma. To approximate the HEK293 cell morphology, we set the soma diameter and length to 40 μm , based on typical HEK293 cell dimensions. Since HEK293 cells lack dendritic processes like neurons, we modeled the cell as a single-compartment soma, without dendritic compartments.

Furthermore, our cell models do not include any ion channels other than KCNQ or KCNQ_{inactivation}, as our goal was to examine how KCNQ channels alone respond to current and voltage changes. In the present study, we examined properties of Kcnq2/3 and Kcnh7 channels by heterologous expression using HEK293 cells. Although this experiment is very useful and widely utilized to analyze protein function, there are some issues to be aware of when interpreting the results. All stem cells used for heterologous expression experiments express their own endogenous proteins, including ion channels. Therefore, experiments using heterologous expression systems cannot completely exclude the possibility that these endogenous factors may unexpectedly affect the resonance and oscillation either through direct interactions or functional crosstalk. The same issue would arise if we included additional ion channels in our computational

model, as their interactions with KCNQ/KCNH channels would complicate the interpretation of our findings.

To circumvent this limitation, we used the computational approach, allowing us to isolate the contribution of KCNQ channels in a completely controlled environment. Our computational analyses successfully demonstrated that KCNQ channels alone—without other co-expressed factors—are capable of generating resonance. This finding underscores the importance of computational analyses with minimal additional variables, enabling us to focus exclusively on the target ion channel.

To clarify this rationale, we have now added an explanation for the necessity of computational modeling in the revised manuscript (page 10, line 225–228) as follows: *“HEK293 cells express their own endogenous proteins that include ion channels, which may unexpectedly affect resonance and oscillation by binding to and/or co-operating with KCNQ channels. To estimate influences of these factors, we conducted computational simulations using the NEURON simulator (v8.0)²⁸.”*

Furthermore, we now provide .hoc and .mod files to perform present simulations in NEURON simulator in Supplementary Data 2.

2a) Another important related example: In Fig. 3, it is mentioned that "currents were simulated using LTSpice XVII", without any further detail. This is entirely insufficient to describe a computational experiment. The entirety of the details of the simulation and the simulation environment (case and point, this is the first I've heard of LTSpice) should be included so that they can be replicated.

[Our response to the comment]

We utilized LTSpice XVII (Analog Devices, Norwood, MA, USA, <https://www.analog.com/en/resources/design-tools-and-calculators/ltspice-simulator.html>), a widely

used, free simulation tool for designing and testing analog electrical circuits in engineering. Its intuitive graphical interface allows for easy construction of circuits by simply placing and connecting analog components such as

Figure 1. LTSpice

resistors, capacitors and inductors, and specifying their parameters (Fig. 1 shown in this document). In our simulations, we configured the circuit as depicted in Fig. 3a and analyzed the current responses to step voltage inputs (Fig. 3b, c). We now add specific parameters for these components in the legend of Fig. 3a (page 22, line 435-436).

“a) Resistor-inductor-capacitor (RLC) circuit with a leaky inductor (R_L [30 M Ω] and L [10 MH]) and resistance of the recording electrode (R_e [5 M Ω]). $R = 100$ M Ω . $C = 200$ pF.”

3) As presented, the computational model is used largely to replicate findings already observed experimentally. This does nothing to advance any sort of argument of the paper, and is a very superficial use of computational modeling. While the alteration of the KCNQ channel into an inactivating channel is interesting, the presented results are once again entirely superficial. The computational setting allows for a wealth of additional explorations that would, in fact, be very useful in this manuscript in deciphering the actual mechanism at play relating m-channel activity to resonance, but there is no effort made to utilize these resources.

[Our response to the comment]

We do not consider the computational analysis in this study to be merely a replication of experimentally obtained data. Instead, computational experiments serve as a powerful tool to test hypotheses that cannot be fully addressed by biological experiments alone. As discussed in your 2nd comment, heterologous expression systems, such as HEK293 cells, may introduce confounding effects due to the presence of endogenous factors that may interact with KCNQ and KCNH channels, potentially influencing resonance and oscillatory behavior. The fact that our simplified KCNQ model was able to generate resonance strongly indicates that KCNQ channels can generate resonance, independent of other cellular components.

Furthermore, in the previous version of our manuscript, the computational analysis presented in Fig. 6 provided the sole direct evidence demonstrating that rapid ion channel inactivation plays a crucial role in inductor-like behavior. While our experimental data comparing Kcnq2/3 and Kcnh7 channel activities led us to speculate about the importance of rapid inactivation, we were unable to definitively conclude this due to the lack of direct experimental verification. To address this, we incorporated rapid inactivation into our computational KCNQ model, allowing us to systematically assess its impact. We firmly believe that computational modeling is a methodologically appropriate approach when direct experimental validation is challenging or unfeasible.

To further strengthen our conclusions about the role of rapid inactivation, we have now included additional biological experiments in the revised manuscript (Supplementary Fig. 2). We recently knew ICA105574, the drug that can block the rapid inactivation of KCNH channels (Zhou *et al.*, 2011). As predicted by our computational analysis, ICA105574 effectively abolished the inductor-like behavior of Kcnh7, providing strong experimental confirmation of our model's predictions.

Taken together, we believe that computational and cell biological approaches complemented each other to produce great advances in this research. This new pharmacological data has been included in Supplementary Fig. 2 and is now described in the Results section (page 12, line 263–269) and discussed in the Discussion section (page 16, line 366–368).

“Finally, we experimentally confirmed the crucial role of the rapid inactivation on Kcnh7 channel gating. Rapid inactivation of Kcnh7 channels was reported to be suppressed by bath application of ICA-105574 (2 μ M)³⁵. As predicted by computational analyses, sudden current changes at the onset and offset of square voltage steps emerged (Supplementary Fig. 2a, b), and opposing conductance changes were suppressed (Supplementary Fig. 2c–f) in Kcnh7-expressing cells treated with ICA-105574. These data support the computational analysis conclusion that the rapid inactivation and recovery dynamics are essential for the opposing conductance change of Kcnh7 channels.”

“In the present study, the computational and experimental analyses revealed that the opposing conductance change arises from the rapid inactivation and recovery from this inactivation of ion channels (Figs. 6c, d and Supplementary Fig. 2).”

As the manuscript currently stands, these two criticisms are, unfortunately, quite serious. When there are no underlying scientific issues, but rather issues with presentation and/or argumentation, my philosophy is usually to defer to the author and provide an opportunity to address these concerns in a revision. I believe that a thorough revision could certainly address my first major concern in order to clearly delineate the hypothesis motivating this work and how it advances our understanding of resonance in neurons, rather than the current version that is overly descriptive. However, I do have significant worries about how the computational component of the paper integrates with the experimental component: in order for the computational element to serve a vital role in the paper's argument rather than simply feel "tacked on", more computational experiments would certainly be needed that probe the relationship between the m-

current and resonance in a manner that complements the experimental argument, rather than merely replicating it. As such, I imagine that the final decision on this manuscript will come down to the opinions of my fellow reviewers regarding the experimental component of the paper, which is the majority and therefore should be weighed more significantly than my concerns about the computational element.

[Our response to the comment]

As we explained in the second part of your concerns, our computational analysis was conducted to address questions that cannot be resolved through cell biological experiments alone. Our simulations demonstrated that KCNQ and KCNQ_{inactivation} channels have the intrinsic capacity to generate resonance without requiring additional co-factors. Moreover, computational modeling allowed us to directly investigate the role of inactivation in resonance—an issue that had remained unresolved in our previous research. While space constraints limited the number of computational experiments we could include, there is no doubt that these analyses have significantly advanced our understanding of resonance.

While I'm disappointed I cannot be more positive in my review at this time, I want to applaud the authors for what was clearly a significant undertaking, as well as for studying what I believe to be a very interesting (and perhaps understudied) question in the field. It's my sincere hope that these comments will lead to an improved manuscript regardless of its final home.

Reviewer #3 (Remarks to the Author):

The paper discuss the ability of specific ion channels to create resonance, only have some frequencies pass the membrane, the exact mechanism in discussed. The paper and the results, are interesting, and well conducted.

I will suggest to expand the NEURON model a bit more, the experimental results showed many possible mechanisms and some were suggested as responsible. And different KO experiments. The model can be extended to show replication and alignment of the theories. The modeling work required is not intense, as this is a single neuron, I suggest to create 1/2 notebook in which you replicate more experiments and show the results. Currently no analysis/model code is supplied it should extended be part of the paper.

Thank you.

[Our response to the comment]

Thank you for your positive evaluation of our research. Unfortunately, we were not entirely sure what you meant by "1/2 notebook", but we interpreted it as a reference to the code necessary to reproduce and expand our simulations. To ensure reproducibility, we have now provided .hoc and .mod files to perform present simulations in NEURON simulator in Supplementary Data 2.

References

- Abidi A, Devaux JJ, Molinari F, Alcaraz G, Michon FX, Sutera-Sardo J, Becq H, Lacoste C, Altuzarra C, Afenjar A, Mignot C, Doummar D, Isidor B, Guyen SN, Colin E, De La Vaissiere S, Haye D, Trauffler A, Badens C, Prieur F, Lesca G, Villard L, Milh M & Aniksztejn L. (2015). A recurrent KCNQ2 pore mutation causing early onset epileptic encephalopathy has a moderate effect on M current but alters subcellular localization of Kv7 channels. *Neurobiol Dis* **80**, 80-92.
- Brown DA & Adams PR. (1980). Muscarinic suppression of a novel voltage-sensitive K⁺ current in a vertebrate neurone. *Nature* **283**, 673-676.
- Carnevale NT & Hines ML. (2006). *The NEURON book*. Cambridge University Press, Cambridge, UK.
- Hashimoto K. (2019). Mechanisms for the resonant property in rodent neurons. *Neurosci Res*.
- Hutcheon B & Yarom Y. (2000). Resonance, oscillation and the intrinsic frequency preferences of neurons. *Trends Neurosci* **23**, 216-222.
- Matsumoto-Makidono Y, Nakayama H, Yamasaki M, Miyazaki T, Kobayashi K, Watanabe M, Kano M, Sakimura K & Hashimoto K. (2016). Ionic Basis for Membrane Potential Resonance in Neurons of the Inferior Olive. *Cell reports* **16**, 994-1004.
- Matsuoka T, Yamasaki M, Abe M, Matsuda Y, Morino H, Kawakami H, Sakimura K, Watanabe M & Hashimoto K. (2021). Kv11 (ether-a-go-go-related gene) voltage-dependent K(+) channels promote resonance and oscillation of subthreshold membrane potentials. *J Physiol* **599**, 547-569.
- Miceli F, Millevert C, Soldovieri MV, Mosca I, Ambrosino P, Carotenuto L, Schrader D, Lee HK, Riviello J, Hong W, Risen S, Emrick L, Amin H, Ville D, Edery P, de Bellescize J, Michaud V, Van-Gils J, Goizet C, Willemsen MH, Kleefstra T, Moller RS, Bayat A, Devinsky O, Sands T, Korenke GC, Kluger G, Mefford HC, Brilstra E, Lesca G, Milh M, Cooper EC, Tagliatela M & Weckhuysen S. (2022). KCNQ2 R144 variants cause neurodevelopmental disability with language impairment and autistic features without neonatal seizures through a gain-of-function mechanism. *EBioMedicine* **81**, 104130.

- Miceli F, Soldovieri MV, Ambrosino P, Barrese V, Migliore M, Cilio MR & Tagliatela M. (2013). Genotype-phenotype correlations in neonatal epilepsies caused by mutations in the voltage sensor of K(v)7.2 potassium channel subunits. *Proc Natl Acad Sci U S A* **110**, 4386-4391.
- Miceli F, Soldovieri MV, Ambrosino P, De Maria M, Migliore M, Migliore R & Tagliatela M. (2015). Early-onset epileptic encephalopathy caused by gain-of-function mutations in the voltage sensor of Kv7.2 and Kv7.3 potassium channel subunits. *J Neurosci* **35**, 3782-3793.
- Varghese N, Moscoso B, Chavez A, Springer K, Ortiz E, Soh H, Santaniello S, Maheshwari A & Tzingounis AV. (2023). KCNQ2/3 Gain-of-Function Variants and Cell Excitability: Differential Effects in CA1 versus L2/3 Pyramidal Neurons. *J Neurosci* **43**, 6479-6494.
- Zhou PZ, Babcock J, Liu LQ, Li M & Gao ZB. (2011). Activation of human ether-a-go-go related gene (hERG) potassium channels by small molecules. *Acta Pharmacol Sin* **32**, 781-788.

Reviewer #1 (Remarks to the Author):

I thank the authors for carefully implementing the feedback for better clarity of the presentation of the data. All the issues raised by me are now resolved and I consider the article suitable for publication.

Reviewer #2 (Remarks to the Author):

I applaud the authors for the effort and care that they've taken in the revision process, yielding a significantly improved manuscript. In particular, I found the new text more precisely defining the "inductor-like" terminology to be of great use to the interpretability of the paper's overall argument, and the contextualization of the computational results is also much more apparent in the revised manuscript.

There are, however, some serious concerns that I raised in my initial review that the authors did not address in their revision. While these decisions were explained in the "rebuttal" document, I unfortunately found such explanations insufficient for the three specific, important points I outline below. It is my opinion that these points need to be fully addressed in the manuscript not only for scientific rigor, but for it to meet the stated standards for publication in *Communications Biology*, particularly regarding this work's ability to influence thinking in the larger field of neurophysiology.

1) In order for this manuscript to be broadly impactful, rather than just of interest to those who study KCNQ and KCNH channels, the findings must be properly contextualized relative to the entire literature relating to subthreshold resonance. The authors state in their rebuttal that, "It has been assumed that the resonant property of the cell membrane is generated by ion channels acting as inductors" without any citations or justification. I, in fact, would strongly disagree with this assertion, and posit that there are multiple proposed mechanisms by which subthreshold resonance can arise in neurons through the activity of differing ion channels. At bare minimum, statements of this type need to be more fully justified in the manuscript, and my strong suggestion is that the authors fully describe the multiple ways in which this dynamic can arise in neurons (including the T-type channel, which they mention in the manuscript, as well as the h-channel, which I raised in my previous review with citations) and how their findings fit in this larger literature.

[Our response to the comment]

We believe it is important to discuss candidate ion channels and their underlying mechanisms separately. As the reviewer pointed out, a number of ion channels have been implicated in contributing to membrane resonance, including HCN, Kcnh, Kcnq channels, T-type voltage-dependent Ca²⁺ channels (VDCCs), and persistent Na⁺ channels (Hutcheon & Yarom, 2000; Hashimoto, 2020). However, the fact that multiple channel types are involved in resonance does not necessarily imply that they each operate through entirely distinct mechanisms. As proposed by Hutcheon and Yarom in their seminal review, these channels can be largely categorized into two functional groups based on their contributions to resonance: resonating conductance and amplifying conductance (Hutcheon & Yarom, 2000).

Ion channels classified as resonating conductances have been proposed to share a key property: their behavior mimics that of an electrical inductor (Hutcheon & Yarom, 2000; Hashimoto, 2020). This inductor-like property allows them to form a parallel resonant circuit in conjunction with the intrinsic membrane resistance and capacitance, ultimately giving rise to resonance. HCN, Kcnq, and Kcnh channels are consistently reported to act as resonating conductance. In contrast, amplifying conductance enhances the activity of resonating conductance but do not themselves directly promote resonance (Hutcheon & Yarom, 2000; Hashimoto, 2020). Persistent Na⁺ channels are widely regarded as typical amplifying conductance. T-type VDCCs have been reported to function as either resonating (Hutcheon *et al.*, 1994; Hutcheon & Yarom, 2000) or amplifying conductances (Matsumoto-Makidono *et al.*, 2016).

To the best of our knowledge, many studies investigating neuronal resonance are grounded on this conceptual framework. In fact, the report by Kalmbach *et al.*—that referenced by the reviewer in the second comment—also explicitly refer to HCN channels as “phenomenological inductance” (Kalmbach *et al.*, 2018). The Hutcheon and Yarom framework has been a widely accepted for understanding subthreshold membrane resonance, which can explain many aspects of experimental data.

Our study also builds on this conceptual model, rather than evaluating each candidate ion channel functions. Notably, we extend the original Hutcheon and Yarom framework by proposing a refinement: ion channels that fall under the category of resonating conductances may be subdivided into two functionally distinct subgroups, based on differences in their gating properties.

In line with the framework established by Hutcheon & Yarom, we have revised the Introduction and Discussion sections of the manuscript to incorporate a clearer explanation of the functional roles of each ion channel (i.e., whether they serve as resonating or amplifying conductances).

(page 3, line 53 – page 4, line 65)

“In living cells, the inductive component is proposed to be mediated by ion channels capable of mimicking inductor-like behavior, which is called resonating conductance¹⁵. Ion channels consistently reported to contribute resonating conductance include hyperpolarization-activated cyclic nucleotide-gated (HCN) channels and KCNH channels^{14, 15}. In addition to resonating conductance, resonance also depends on amplifying conductance, which boosts resonating conductance activity in a voltage-dependent manner^{14, 15}. Persistent sodium channels are reported as amplifying conductance. T-type voltage-dependent calcium channels (VDCCs) have been reported to function as either resonating^{15, 16} or amplifying conductance¹⁷. In some neurons, M-current blockers inhibit resonance at relatively depolarized membrane potentials¹⁸⁻²⁴, and resonance in CA1 pyramidal neurons is impaired in mice expressing a KCNQ2 subunit with a dominant-negative pore mutation²³. These reports suggest that KCNQ channels can function as resonating conductance, particularly at depolarized potentials¹⁴. However, the precise mechanism by which they emulate inductor-like properties to drive resonance remains unclear.”

(page 15, line 352 – page 16, line 372)

“Previous studies have consistently indicated that M-channels (Kcnq), Kcnh channels, and HCN channels possess the potential to function as resonating conductance^{14, 15}. Among these, our findings suggest that the underlying mechanisms by which Kcnq and Kcnh channels contribute to resonance are partially different. Our results indicate that resonating conductance may be functionally subdivided into at least two categories—those mediated by ion channels that exhibit both the opposing conductance change and subsequent slow activation or deactivation (as seen with Kcnh7 channels), and those that exhibit only the slow activation or deactivation (as observed with Kcnq2/3 channels). Meanwhile, the functional classification of HCN channels as resonating conductance remains unresolved. HCN channels have been widely reported to mediate resonance across numerous brain regions^{14, 18, 19, 21, 22, 42-44}. Among four HCN subtypes, contribution of HCN1 has been studied using knockout mice^{17, 45, 46}. Voltage responses of HCN-

*dependent resonance typically exhibit a phase lead relative to chirp current input at the lower frequency range, which suggests that the HCN channel functions as a phenomenological inductor^{17, 44, 47, 48}. However, in our previous study, heterologous expression of HCN1 in HEK293 cells failed to elicit clear membrane potential oscillations or resonance²⁵. Consequently, analyses to test the inductor-like gating property (Fig. 3) have not been performed for HCN1. It thus remains uncertain whether HCN1 channels emulate *Kcnh7*-type or *Kcnq2/3*-type inductor-like performance. Given that HCN channels are subject to extensive modulation by intracellular signaling pathways and auxiliary subunits in neurons⁴⁹, it may be possible that certain neuron-specific interacting partners are required for HCN1 to exhibit inductor-like behavior. Future studies incorporating such factors will be necessary to clarify the precise gating mechanisms underlying HCN1-mediated resonance.”*

2) The statements made by the authors in their rebuttal document regarding the relationship between the h-channel and subthreshold resonance are in my view inaccurate. The work I mentioned in my initial review includes analysis showing that the dynamics of the h-channel indeed are sufficient to drive subthreshold resonance in itself. In fact, the computational strategy taken by Kalmbach et. al. to show this closely parallels the strategy the authors took to make a similar assertion for KCNQ channels in this manuscript. If the authors wish to (reasonably, in my opinion) use their computational results to show that KCNQ channels alone are sufficient to produce subthreshold resonance, they cannot also disregard similar results from Kalmbach et. al. and the related studies I initially mentioned. The reality that the h-channel can induce subthreshold resonance, and how this does or does not support the authors assertions regarding the relationship between inductor-like activity and subthreshold resonance, is something that I believe must be addressed in the Discussion of the manuscript to fully and properly contextualize these results relative to contemporary literature, particularly given the similarities in the computational strategies that I describe above.

[Our response to the comment]

In the reviewer’s original comment, it was noted that selecting *Kcnh7* as the representative ion channel exhibiting inductor-like behavior could be viewed as somewhat arbitrary. In response to this, we presented ion channels that have been consistently reported to generate resonance and explained why other major candidate channels—particularly HCN channels—were not used as direct comparisons in our study. However, we would like to clarify that our intention was not to downplay the essential

role of HCN channels in generating neuronal resonance.

We fully agree with the reviewer's view that HCN channels, particularly *HCN1* (Nolan *et al.*, 2007; Giocomo & Hasselmo, 2009; Matsumoto-Makidono *et al.*, 2016), play a critical role in generating resonance in neurons. This conclusion is strongly supported by numerous experimental studies, including the work by Kalmbach *et al.* (Hutcheon *et al.*, 1996; Hu *et al.*, 2002; Ulrich, 2002; Tanaka *et al.*, 2003; Nguyen *et al.*, 2004; Wang *et al.*, 2006; Haas *et al.*, 2007; Manuel *et al.*, 2007; Narayanan & Johnston, 2007, 2008; Hu *et al.*, 2009; Marcelin *et al.*, 2009; Dembrow *et al.*, 2010; Heys *et al.*, 2010; Zemankovics *et al.*, 2010; Gastrein *et al.*, 2011; Dwyer *et al.*, 2012; Marcelin *et al.*, 2012; Shay *et al.*, 2012; Sun *et al.*, 2012; Xue *et al.*, 2012; Yan *et al.*, 2012; Boehlen *et al.*, 2013; Ulrich, 2014; Vera *et al.*, 2014; Yang *et al.*, 2015; Hu *et al.*, 2016; Kalmbach *et al.*, 2018), and aligns with our own findings using inferior olivary neurons (Matsumoto-Makidono *et al.*, 2016). In neurons, HCN-dependent resonance exhibits a phase lead relative to sinusoidal current inputs at lower frequencies—an electrophysiological signature that has been interpreted as evidence of inductor-like behavior (Ulrich, 2002; Narayanan & Johnston, 2008; Marcelin *et al.*, 2012; Matsumoto-Makidono *et al.*, 2016).

However, in contrast to what would be expected from these neuronal findings, our previous study showed that expression of *HCN1* alone in HEK293 cells did not produce clear resonance or membrane potential oscillations (Matsuoka *et al.*, 2021). This suggests that *HCN1* by itself may be insufficient to elicit a robust frequency-dependent response in a non-neuronal context. We speculate that essential modulatory components—present in neurons but absent in HEK293 cells—may be necessary for *HCN1* to function as a reliable resonating conductance.

Indeed, HCN channel function in neurons is known to be heavily influenced by a range of interacting proteins and regulatory molecules, such as *PEX5L*, *KCNE2*, *IRAG1*, *FLNA*, and cyclic nucleotides (Peters *et al.*, 2022). Because of this complexity, it remains experimentally challenging to determine whether *HCN1* alone is sufficient to induce resonance in neurons. This limitation also extends to computational models based on neuronal parameters, which may inadvertently incorporate effects of such modulatory factors without being able to isolate the intrinsic properties of the channel itself.

Due to the lack of pronounced resonance and oscillatory behavior in *HCN1*-expressing HEK293 cells, we were unable to perform the same level of detailed electrophysiological

analysis that were conducted for *Kcnh7*- and *Kcnq2/3*-expressing cells (Fig. 3 and 4). Consequently, it currently remains unclear whether *HCN1* exhibits the inductor-like (as observed for *Kcnh7*) or more moderate, limited-inductor-like (as seen with *Kcnq2/3*) properties.

According to the reviewer's suggestion, we have now added these discussions to the Discussion section of the revised manuscript to more clearly articulate the rationale behind our experimental design and to better contextualize the role of HCN channels in the broader framework of resonance mechanisms (page 15, line 352 – page 16, line 372).

*“Previous studies have consistently indicated that M-channels (*Kcnq*), *Kcnh* channels, and HCN channels possess the potential to function as resonating conductance^{14, 15}. Among these, our findings suggest that the underlying mechanisms by which *Kcnq* and *Kcnh* channels contribute to resonance are partially different. Our results indicate that resonating conductance may be functionally subdivided into at least two categories—those mediated by ion channels that exhibit both the opposing conductance change and subsequent slow activation or deactivation (as seen with *Kcnh7* channels), and those that exhibit only the slow activation or deactivation (as observed with *Kcnq2/3* channels). Meanwhile, the functional classification of HCN channels as resonating conductance remains unresolved. HCN channels have been widely reported to mediate resonance across numerous brain regions^{14, 18, 19, 21, 22, 42-44}. Among four HCN subtypes, contribution of *HCN1* has been studied using knockout mice^{17, 45, 46}. Voltage responses of HCN-dependent resonance typically exhibit a phase lead relative to chirp current input at the lower frequency range, which suggests that the HCN channel functions as a phenomenological inductor^{17, 44, 47, 48}. However, in our previous study, heterologous expression of *HCN1* in HEK293 cells failed to elicit clear membrane potential oscillations or resonance²⁵. Consequently, analyses to test the inductor-like gating property (Fig. 3) have not been performed for *HCN1*. It thus remains uncertain whether *HCN1* channels emulate *Kcnh7*-type or *Kcnq2/3*-type inductor-like performance. Given that HCN channels are subject to extensive modulation by intracellular signaling pathways and auxiliary subunits in neurons⁴⁹, it may be possible that certain neuron-specific interacting partners are required for *HCN1* to exhibit inductor-like behavior. Future studies incorporating such factors will be necessary to clarify the precise gating mechanisms underlying *HCN1*-mediated resonance.”*

3) The revision to the text the authors made in response to point 3 in my original review

misses the point. The new text references "the responsiveness of neurons to periodic inputs near the resting membrane potential, which might lead to abnormal increases in synchronized network activity." However, the literature, particularly the citations I referenced previously, do not support this statement: there is not necessarily a one-to-one relationship between the sub- and supra-threshold frequency preferences of neurons, particularly as characterized by the response to a ZAP input. As another example, the statement on Lines 50-52 that, "Resonance is thought to be crucial for synchronized activity of multiple neurons across networks and for generating cell-autonomous oscillations at characteristic frequencies" is not inherently true given the non uniform relationship between sub- and supra-threshold resonance. Properly contextualizing their results relative to this distinction is necessary for this work to be of interest to the broader field. While my opinion is that the results presented here are insufficient to make any claims about synchronized network activity at the level of neuronal spikes, please note this isn't a criticism of the quality or impact of the work at all; rather, as initially mentioned, this is simply a caution to not unnecessarily "oversell" the results. I would strongly encourage the authors review the citations I included in my original review relative to this point and adjust their presentation of their results accordingly.

[Our response to the comment]

Both *Kcnq* and *Kcnh* channels are activated by depolarizations from the resting membrane potential (approximately -50 mV; Fig. 1b), and their resonant responses become evident within the depolarized range of -40 to -20 mV (Fig. 1k, l) (Matsuoka *et al.*, 2021). Therefore, *Kcnq* and *Kcnh* channels primarily contribute to “supra-threshold resonance”, if “supra-threshold” means the membrane potential above the action potential threshold. While the reviewer's point is very important for considering roles of resonance, we respectfully consider that a detailed discussion on the interplay between supra-threshold frequency preference and sub-threshold resonance—typically mediated by HCN channels—extends beyond the specific scope of this study.

Resonance mediated by ion channels at depolarized membrane potentials has been reported in several studies. In such contexts, the underlying conductances often alter between depolarized and hyperpolarized membrane potential (Hutcheon *et al.*, 1996; Hu *et al.*, 2002; Hu *et al.*, 2009; Dembrow *et al.*, 2010; Heys *et al.*, 2010; Boehlen *et al.*, 2013; Vera *et al.*, 2014; Yang *et al.*, 2015; Matsumoto-Makidono *et al.*, 2016; Matsuoka *et al.*, 2021). Resonance at depolarized potential is primarily governed by *Kcnq*, *Kcnh*, and persistent sodium channels, whereas hyperpolarization-activated resonance is typically mediated by *HCN* channels.

The supra-threshold resonance may affect peri- to supra-threshold frequency preference. With respect to *Kcnq* channels, a previous study has shown that loss-of-function mutations abolish theta-range (6–10 Hz) rhythmicity and resonance in the hippocampus (Peters *et al.*, 2005). Furthermore, there is a computational analysis suggesting that the M-current promotes the precise coordination of gamma frequency spike firings in the presence of the theta range activity (Zhou *et al.*, 2018). These data underscore the critical role of the *Kcnq* conductance in maintaining proper rhythmic activity in the brain.

In contrast, the contribution of *Kcnh* channels to network dynamics remains poorly understood, because their function as resonant conductances has only recently been recognized (Matsuoka *et al.*, 2021). Clarifying their impact on network-level oscillatory activity will require further investigation.

However, as noted by the reviewer, our current dataset—derived exclusively from heterologous expression experiments—is not sufficient to directly address the influence of *Kcnq*-mediated resonance on neural network function. In consideration of this limitation, we have decided to omit lines 50–52 from the previous revision, and to revise the Discussion section (page 16, lines 378 – page 17, line 381) accordingly to more accurately reflect the limitations of our approach.

“Notably, the hyperpolarizing shift of resonance induced by the R144Q GOF mutant could enhance the responsiveness of neurons to periodic inputs near the resting membrane potential. This might result in abnormal frequency response of individual cells and contribute to pathological states.”

While I feel strongly about the necessity for these three points to be fully addressed, I also acknowledge that my assessment is somewhat of an outlier relative to the other two peer reviewers. Thus, I am comfortable letting the editor decide how significant the revisions to address the above points need to be and whether they can be made as a "minor revision" or whether another round of peer review is necessary. If it is the later, I would ask the authors include a "tracked changes" version of the manuscript with their resubmission, as is typically required, to highlight new and revised text in the manuscript.

References

- Boehlen A, Henneberger C, Heinemann U & Erchova I. (2013). Contribution of near-threshold currents to intrinsic oscillatory activity in rat medial entorhinal cortex layer II stellate cells. *J Neurophysiol* **109**, 445-463.
- Dembrow NC, Chitwood RA & Johnston D. (2010). Projection-specific neuromodulation of medial prefrontal cortex neurons. *J Neurosci* **30**, 16922-16937.
- Dwyer J, Lee H, Martell A & van Drongelen W. (2012). Resonance in neocortical neurons and networks. *Eur J Neurosci* **36**, 3698-3708.
- Gastrein P, Campanac E, Gasselín C, Cudmore RH, Bialowas A, Carlier E, Fronzaroli-Molinieres L, Anki N & Debanne D. (2011). The role of hyperpolarization-activated cationic current in spike-time precision and intrinsic resonance in cortical neurons in vitro. *J Physiol* **589**, 3753-3773.
- Giocomo LM & Hasselmo ME. (2009). Knock-out of HCN1 subunit flattens dorsal-ventral frequency gradient of medial entorhinal neurons in adult mice. *J Neurosci* **29**, 7625-7630.
- Haas JS, Dorval AD, 2nd & White JA. (2007). Contributions of Ih to feature selectivity in layer II stellate cells of the entorhinal cortex. *J Comput Neurosci* **22**, 161-171.
- Hashimoto K. (2020). Mechanisms for the resonant property in rodent neurons. *Neurosci Res* **156**, 5-13.
- Heys JG, Giocomo LM & Hasselmo ME. (2010). Cholinergic modulation of the resonance properties of stellate cells in layer II of medial entorhinal cortex. *J Neurophysiol* **104**, 258-270.
- Hu H, Vervaeke K, Graham LJ & Storm JF. (2009). Complementary theta resonance filtering by two spatially segregated mechanisms in CA1 hippocampal pyramidal neurons. *J Neurosci* **29**, 14472-14483.
- Hu H, Vervaeke K & Storm JF. (2002). Two forms of electrical resonance at theta frequencies, generated by M-current, h-current and persistent Na⁺ current in rat hippocampal pyramidal cells. *J Physiol* **545**, 783-805.

- Hu R, Ferguson KA, Whiteus CB, Meijer DH & Araneda RC. (2016). Hyperpolarization-Activated Currents and Subthreshold Resonance in Granule Cells of the Olfactory Bulb. *eNeuro* **3**.
- Hutcheon B, Miura RM & Puil E. (1996). Subthreshold membrane resonance in neocortical neurons. *J Neurophysiol* **76**, 683-697.
- Hutcheon B, Miura RM, Yarom Y & Puil E. (1994). Low-threshold calcium current and resonance in thalamic neurons: a model of frequency preference. *J Neurophysiol* **71**, 583-594.
- Hutcheon B & Yarom Y. (2000). Resonance, oscillation and the intrinsic frequency preferences of neurons. *Trends Neurosci* **23**, 216-222.
- Kalmbach BE, Buchin A, Long B, Close J, Nandi A, Miller JA, Bakken TE, Hodge RD, Chong P, de Frates R, Dai K, Maltzer Z, Nicovich PR, Keene CD, Silbergeld DL, Gwinn RP, Cobbs C, Ko AL, Ojemann JG, Koch C, Anastassiou CA, Lein ES & Ting JT. (2018). h-Channels Contribute to Divergent Intrinsic Membrane Properties of Supragranular Pyramidal Neurons in Human versus Mouse Cerebral Cortex. *Neuron* **100**, 1194-1208 e1195.
- Manuel M, Meunier C, Donnet M & Zytnicki D. (2007). Resonant or not, two amplification modes of proprioceptive inputs by persistent inward currents in spinal motoneurons. *J Neurosci* **27**, 12977-12988.
- Marcelin B, Chauviere L, Becker A, Migliore M, Esclapez M & Bernard C. (2009). h channel-dependent deficit of theta oscillation resonance and phase shift in temporal lobe epilepsy. *Neurobiol Dis* **33**, 436-447.
- Marcelin B, Liu Z, Chen Y, Lewis AS, Becker A, McClelland S, Chetkovich DM, Migliore M, Baram TZ, Esclapez M & Bernard C. (2012). Dorsoventral differences in intrinsic properties in developing CA1 pyramidal cells. *J Neurosci* **32**, 3736-3747.
- Matsumoto-Makidono Y, Nakayama H, Yamasaki M, Miyazaki T, Kobayashi K, Watanabe M, Kano M, Sakimura K & Hashimoto K. (2016). Ionic Basis for Membrane Potential Resonance in Neurons of the Inferior Olive. *Cell reports* **16**, 994-1004.
- Matsuoka T, Yamasaki M, Abe M, Matsuda Y, Morino H, Kawakami H, Sakimura K, Watanabe M &

- Hashimoto K. (2021). Kv11 (ether-a-go-go-related gene) voltage-dependent K(+) channels promote resonance and oscillation of subthreshold membrane potentials. *J Physiol* **599**, 547-569.
- Narayanan R & Johnston D. (2007). Long-term potentiation in rat hippocampal neurons is accompanied by spatially widespread changes in intrinsic oscillatory dynamics and excitability. *Neuron* **56**, 1061-1075.
- Narayanan R & Johnston D. (2008). The h channel mediates location dependence and plasticity of intrinsic phase response in rat hippocampal neurons. *J Neurosci* **28**, 5846-5860.
- Nguyen QT, Wessel R & Kleinfeld D. (2004). Developmental regulation of active and passive membrane properties in rat vibrissa motoneurons. *J Physiol* **556**, 203-219.
- Nolan MF, Dudman JT, Dodson PD & Santoro B. (2007). HCN1 channels control resting and active integrative properties of stellate cells from layer II of the entorhinal cortex. *J Neurosci* **27**, 12440-12451.
- Peters CH, Singh RK, Bankston JR & Proenza C. (2022). Regulation of HCN Channels by Protein Interactions. *Front Physiol* **13**, 928507.
- Peters HC, Hu H, Pongs O, Storm JF & Isbrandt D. (2005). Conditional transgenic suppression of M channels in mouse brain reveals functions in neuronal excitability, resonance and behavior. *Nat Neurosci* **8**, 51-60.
- Shay CF, Boardman IS, James NM & Hasselmo ME. (2012). Voltage dependence of subthreshold resonance frequency in layer II of medial entorhinal cortex. *Hippocampus* **22**, 1733-1749.
- Sun H, Luhmann HJ & Kilb W. (2012). Resonance properties of different neuronal populations in the immature mouse neocortex. *Eur J Neurosci* **36**, 2753-2762.
- Tanaka S, Wu N, Hsiao CF, Turman J, Jr. & Chandler SH. (2003). Development of inward rectification and control of membrane excitability in mesencephalic v neurons. *J Neurophysiol* **89**, 1288-1298.
- Ulrich D. (2002). Dendritic resonance in rat neocortical pyramidal cells. *J Neurophysiol* **87**, 2753-

2759.

- Ulrich D. (2014). Subthreshold delta-frequency resonance in thalamic reticular neurons. *Eur J Neurosci* **40**, 2600-2607.
- Vera J, Pezzoli M, Pereira U, Bacigalupo J & Sanhueza M. (2014). Electrical resonance in the theta frequency range in olfactory amygdala neurons. *PLoS One* **9**, e85826.
- Wang WT, Wan YH, Zhu JL, Lei GS, Wang YY, Zhang P & Hu SJ. (2006). Theta-frequency membrane resonance and its ionic mechanisms in rat subicular pyramidal neurons. *Neuroscience* **140**, 45-55.
- Xue WN, Wang Y, He SM, Wang XL, Zhu JL & Gao GD. (2012). SK- and h-current contribute to the generation of theta-like resonance of rat substantia nigra pars compacta dopaminergic neurons at hyperpolarized membrane potentials. *Brain Struct Funct* **217**, 379-394.
- Yan ZQ, Liu SM, Li J, Wang Y, Gao L, Xie RG, Xue WN, Zhang GL, Zhu JL & Gao GD. (2012). Membrane resonance and its ionic mechanisms in rat subthalamic nucleus neurons. *Neurosci Lett* **506**, 160-165.
- Yang J, Hu S, Li F & Xing J. (2015). Resonance characteristic and its ionic basis of rat mesencephalic trigeminal neurons. *Brain Res* **1596**, 1-12.
- Zemankovics R, Kali S, Paulsen O, Freund TF & Hajos N. (2010). Differences in subthreshold resonance of hippocampal pyramidal cells and interneurons: the role of h-current and passive membrane characteristics. *J Physiol* **588**, 2109-2132.
- Zhou Y, Vo T, Rotstein HG, McCarthy MM & Kopell N. (2018). M-Current Expands the Range of Gamma Frequency Inputs to Which a Neuronal Target Entrain. *J Math Neurosci* **8**, 13.